**Analysis**

# Comparing chromatin contact maps at scale: methods and insights

**Ketrin Gjoni** [1,2,7], **Laura M. Gunsalus**[1,2,7], **Shuzhen Kuang** [1,2,7], **Evonne McArthur** [2,3,4,5,7], **Maureen Pittman**[1,2], **John A. Capra** [2,3] ✉ & **Katherine S. Pollard** [1,2,3,6] ✉

Comparing chromatin contact maps is an essential step in quantifying how three-dimensional (3D) genome organization shapes development, evolution, and disease. However, methods often disagree, and no gold standard exists for comparing pairs of maps. Here, we evaluate 25 ways to compare contact maps using Micro-C and Hi-C data from two cell types and in silico-generated contact maps. We identify similarities and differences between the methods and quantify their robustness to common sources of biological and technical variation, including losses and gains of CTCF-binding sites, changes in contact intensity or patterns, and noise. We find that global comparison methods, such as mean squared error, are suitable for initial screening; however, biologically informed methods are necessary for identifying how maps diverge and for proposing specific functional hypotheses. We provide a reference guide, codebase, and thorough evaluation for rapidly comparing chromatin contact maps at scale to enable biological insights into 3D genome organization.

The same genomic locus can adopt different 3D conformations in different cells, species, and disease states, affecting gene regulation, cell identity, and replication timing (Fig. 1a)[1–7]. Chromosome conformation capture methods (3C, 4C, 5C, Hi-C, and Micro-C)[8–12] analyze how the genome folds across scales, including chromosomal territories, topologically associating domains (TADs), enhancer–promoter loops, and architectural stripes[10,13–15]. Recently, single-cell and deep-learning techniques have accelerated the study of chromatin conformations across biological contexts[16–22]. As the volume of data grows, analytical tools for chromatin contact maps are being rapidly developed[23,24]. Many methods aim to detect differences in contact maps for various applications, such as ranking differences between pairs of maps[6,7,25–28], assessing reproducibility between replicates and modalities[7,25,26,29], identifying tissue-specific contacts[28], and highlighting differential chromatin interactions[6,27]. Some of these methods are designed to identify structural differences, such as TADs; others target focal changes,

like loops within them (Fig. 1b). Additionally, there are methods that do not target specific changes. Choosing the appropriate map comparison method for a specific issue requires careful consideration of how different methods prioritize various map features and their sensitivity to technical artifacts. The decision-making process is challenging because methods for contact map comparison have not been benchmarked at scale across diverse use cases.

To address this gap, we developed a unifying framework to guide strategies for comparing contact maps, evaluating 25 commonly used approaches (Supplementary Note). Our benchmark includes statistics for directly comparing matrices (global methods; Fig. 1c, left) and methods for comparing specific biologically motivated summaries of contact maps (contact map methods; Fig. 1c, right). We include four newly adapted contact map methods for analyzing maps predicted by machine-learning models, enabling comparisons in the contexts of in silico screens, synthetic sequence design, and simulations.

[1]Gladstone Institute of Data Science and Biotechnology, San Francisco, CA, USA. [2]Department of Epidemiology & Biostatistics, University of California, San Francisco, CA, USA. [3]Bakar Computational Health Sciences Institute, University of California, San Francisco, CA, USA. [4]Vanderbilt Genetics Institute, Vanderbilt University Medical Center, Nashville, TN, USA. [5]Department of Medicine, University of Washington, Seattle, WA, USA. [6]Chan Zuckerberg Biohub, San Francisco, CA, USA. [7]These authors contributed equally: Ketrin Gjoni, Laura M. Gunsalus, Shuzhen Kuang, Evonne McArthur. ✉e-mail: tony@capralab.org; katherine.pollard@gladstone.ucsf.edu

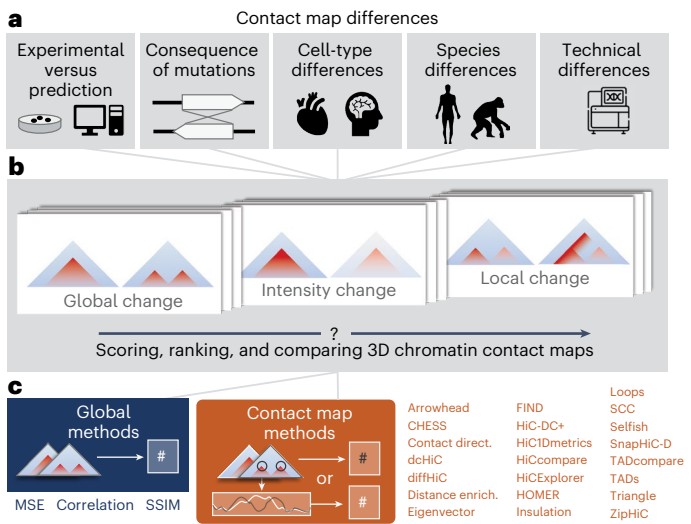

**Fig. 1 | Approaches for comparing 3D chromatin contact maps. a**, 3D genome comparisons drive insights into many domains of chromatin biology. Differences observed between maps might reflect consequences of mutations, cell-type differences, species differences, or technical biases. **b**, 3D contact maps exhibit a range of functionally meaningful differences, for example, in global folding patterns, contact intensity, or small, focal changes in part of the map. **c**, The methods we evaluated include global methods, which are general statistics for comparing matrices, and contact map methods, which are designed to capture specific biologically motivated changes or features of contact maps. The symbol # indicates a numerical score.

To rigorously quantify how different methods score pairs of contact maps, we used three types of data: (1) experimental Micro-C and Hi-C data at different resolutions and map sizes; (2) contact maps predicted by a machine-learning model from DNA sequences, both with and without genetic perturbations, and (3) in silico simulated contact maps that capture specific biological and technical variations. Our analyses reveal instances of divergence and consistency among methods, as well as identify redundant and complementary approaches and the features that they are most sensitive to. We summarize our findings into recommendations and provide a library of open-source code to implement all 25 methods, enabling scientists to choose and apply the right method for their research questions.

## Results

### Diverse strategies for scoring pairs of contact maps

When scoring differences between pairs of contact maps at scale, it is common to compute statistics, such as the mean squared error (MSE) between contact map values, to mathematically compare contact matrices[6,7,20]. These global methods are easy and quick to implement, without embedding specific assumptions about which map features are biologically important. Such statistics are often used as loss functions in models that predict contact maps. It is unclear, however, which method to use. To investigate this, we first compared two widely used global methods, Spearman's correlation coefficient and MSE[30–32], to assess how they rank pairs of Micro-C contact maps from human foreskin fibroblasts (HFFs) and embryonic stem cells (ESCs). Because the Spearman correlation, like several other methods, yields larger values for more similar maps, we transformed all score ranges such that higher values indicate a greater difference in 3D organization (Extended Data Fig. 1), and normalized all scores to a range of 0 to 1 (Methods) to enable comparisons across methods. We refer to this transformed score as 'Correlation.'

Analyzing results across all 7,840 windows, each ~1 Mb ($1 \times 2^{20}$ base pairs (bp)), in the human genome (Methods), we found that Correlation

and MSE often identified differences in markedly different regions (Fig. 2, $r^2 = 0.0002$)[11] for reasons unrelated to the underlying biology. For example, Correlation prioritized a pair of maps exhibiting visible structural rearrangements and low contact frequencies, but MSE did not, because the absolute difference between the maps was small (Fig. 2, top left). Conversely, two maps with similar structures but different contact frequency ranges produced a large MSE, despite being very similar to each other according to Correlation (Fig. 2, bottom right). These inconsistencies are not surprising, given that Correlation is agnostic to intensity changes—consistently increasing values but maintaining their relative levels will not affect Correlation—whereas MSE is sensitive to intensity. This simple example illustrates the importance of considering the types of difference that should be prioritized when selecting a comparison method.

To enhance our understanding, we selected a broad range of global and contact map comparison methods for comprehensive analysis (Fig. 1c, Supplementary Table 1, and Supplementary Note). For global methods, we included Correlation, MSE, and the structural similarity index measure (SSIM). Because these methods overlook specific properties of contact frequency maps, they might struggle to identify smaller, biologically meaningful changes, such as losses and gains of chromatin loops. In these instances, methods that account for expected genome organization structures, which we refer to as contact map methods, could provide an advantage. The first set of these methods—Contact Directionality, Insulation, Distance Enrichment, Eigenvector, and Triangle—transform two-dimensional (2D) contact matrices into one-dimensional (1D) tracks (capturing features relevant to genome folding). These tracks are then compared using Correlation or MSE (indicated with either the (corr) or (mse) suffix, respectively). One benefit of the intermediate 1D tracks is that they can visually evaluate overall map differences along genome coordinates (Extended Data Fig. 2a), although some information present in the full matrix is lost in the reduction to 1D. The second type of contact map method we examine seeks to quantify certain 2D map characteristics, such as changes in loops[31], TADs[33,34], or TAD boundaries. A subset of these methods first calls features and then counts the differences in the number of features to score a pair of maps (Extended Data Fig. 2b).

We evaluate both previously described contact map methods—Arrowhead[31], CHESS[6], dcHiC[35], HiC1Dmetrics[36], HiCcompare[27], Loops[31], stratum-adjusted correlation coefficient (SCC)[7], TADcompare[37], and TADs[34]—as well as four methods—Eigenvector, Contact Directionality, Distance Enrichment, and Triangle—that we implemented on the basis of statistics used in the 3D genome field to assess known properties of contact maps. For each method, we adapted the single-map statistic into a score quantifying the difference in the statistic between a pair of maps. Eigenvector was adapted from compartment-calling methods[38], extending the approach to compare eigenvectors in smaller window sizes. Contact Directionality was adapted from the directionality index[39] and used to evaluate differences in contact directionality, that is whether a region interacts more with up- or downstream regions. Distance Enrichment, adapted from contact decay[9], quantifies differences in contact decay plots between two maps. This method emphasizes how contact patterns between two regions are affected by their distance, focusing on changes in distal contacts. Triangle, a previously used but less-established method[40], calculates average contact frequencies across submatrices and compares these averages between two matrices, ultimately comparing substructures at different levels.

In summary, we describe, implement, and compare 25 methods, including several variants of the same approach, for scoring contact frequency maps. This set covers most commonly used approaches, excluding a small number that could not be implemented within the benchmarking datasets used in this study (Supplementary Table 1).

We assessed the performance of these methods across diverse settings, including maps generated by experiments (Micro-C and HiC), machine-learning models, and direct simulation. We first applied all

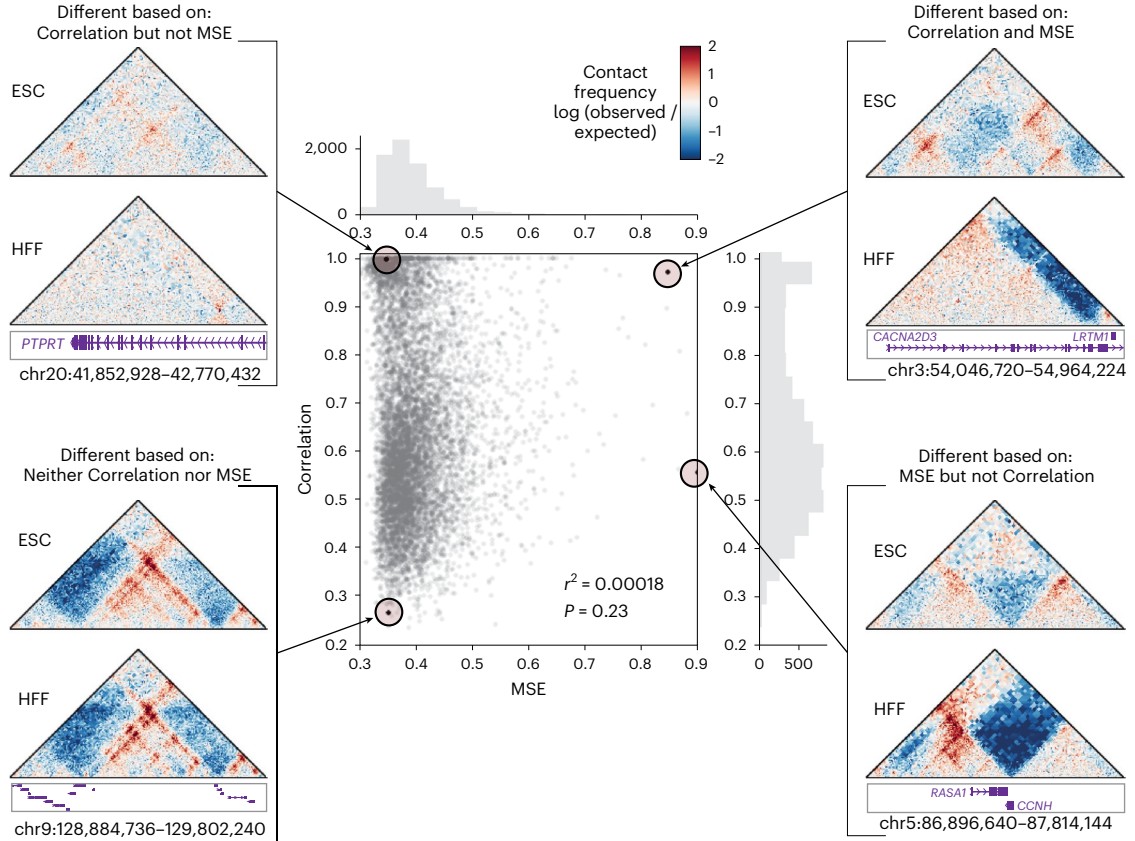

**Fig. 2 | Correlation and MSE score contact frequency map pairs differently.**
Spearman correlation and MSE were calculated across the genome on
experimental contact maps from ESCs and HFFs ($n$ = 7,840). Each point
represents the scores for a pair of contact maps. The histograms on the top and
right summarize the counts of pairs falling across the range of values for MSE
(top) or Correlation (right). To facilitate comparison with MSE, the Correlation
score is a transformed version of the Spearman correlation such that similar
maps receive scores near 0 and different maps receive scores near 1 (Methods).

Shown are examples in which only Correlation prioritizes the map pair as
different (top left), both methods agree the maps are similar (bottom left), both
methods agree the maps are different (top right), and only MSE prioritizes map
pairs as different (bottom right). Genes in these regions are shown in the genomic
tracks below the maps in purple, except for the bottom left map, in which there
are too many genes to include names in the figure. Pearson correlation ($P$) and the
coefficient of determination ($r^2$) are shown in the figure.

25 methods to experimental maps of regions around differentially
expressed genes (DEGs) between HFFs and ESCs, because these regions
are likely to exhibit contact differences. We then evaluated 13 of the
methods applicable to predicted maps in a large-scale screen of in
silico genetic perturbations. Finally, we used simulated predicted maps
to highlight method sensitivities to various technical and biological
variations and to quantitatively measure performance. This extensive
three-step evaluation explores how different methods rank map pairs
to uncover their similarities, unique advantages, and sensitivities.
Finally, we suggest guidelines to help users choose a method or set of
methods catering to their application and goals.

**Evaluation of map comparison methods on experimental data**
To quantify how methods in the representative set compare to each
other, we applied them to Micro-C and Hi-C datasets from ESC and
HFF cell lines, normalized with a commonly used set of preprocessing
steps (Methods). We first conducted a qualitative assessment using
pairs of example HFF versus ESC maps with clear structural changes,
such as variations in TAD boundaries, stripes, or loops (Extended Data
Fig. 2). Eigenvector highlights signal at the example boundary and
stripe differences, but less specifically at loop disparities. However,
Contact Directionality identifies loop changes, suggesting that it is
more sensitive to focal changes. As expected, Distance Enrichment
shows the highest signal for the boundary difference map pair, which
has the most changes in distal contacts. Next, we evaluated the TADs

and Loops methods, which quantify features in each map and generate
an overlap ratio on the basis of those counts. These methods have a
small range of possible scores owing to the low number of features in
many 1-Mb windows. Two example maps with TAD and loop differences
each show three features in ESCs and two features in HFFs, resulting in
an overlap ratio of two-thirds (Extended Data Fig. 2b). Together, these
qualitative results provide insights into the best use-cases for the newly
adapted methods and highlight key challenges associated with apply-
ing count-based methods on a genome-wide scale.

Next, we conducted a large-scale quantitative comparison of
the 25 map comparison scores (Supplementary Table 1) in ESCs and
HFFs at genomic windows around DEGs, where we anticipate various
contact changes (Methods). To discover which methods agree and
disagree in scoring DEG contact maps, we computed clusters and a
correlation matrix and performed principal component analysis (PCA)
(Fig. 3). Because some of the methods have computational limitations,
we focused on chromosomes 21 and 22. For methods that require
parameter tuning, we used visual assessment on a subset of maps
to select parameters that perform well on average (Supplementary
Figs. 1 and 2)[41,42].

Overall, we observe several patterns in how contact maps are
scored by different methods. First, the Correlation and MSE versions
of techniques, such as dcHiC (mse) and dcHiC (corr), tend to cluster
closely with the versions of other methods using the same statistic,
but diverge from one another (Fig. 3), which is not surprising given

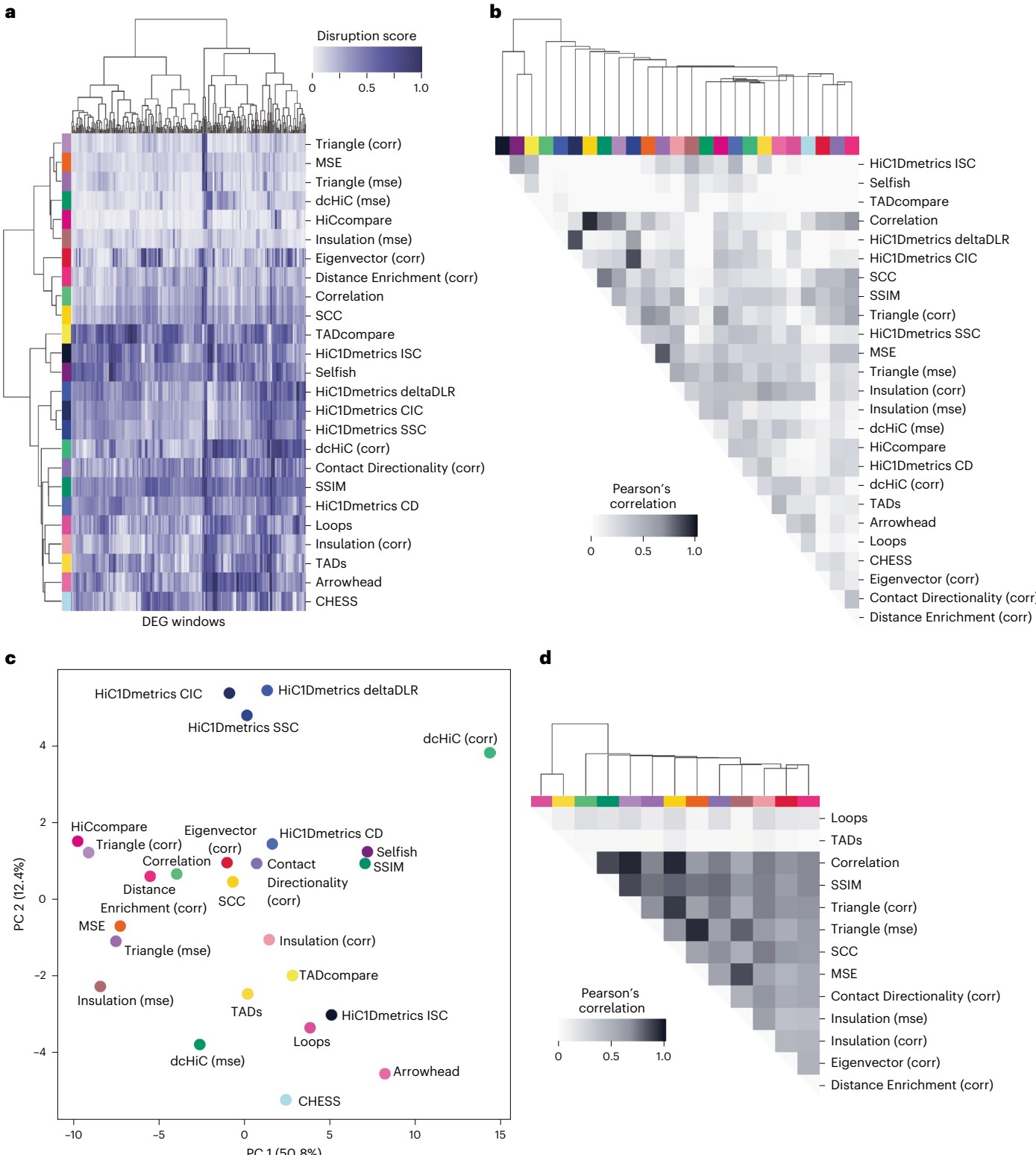

**Fig. 3 | Comparison of methods for evaluating contact frequency maps.**
**a**, A heatmap showing scores from 25 methods across 256 Micro-C maps, which represent 1-Mb regions at 2,048-bp resolution around DEGs between ESCs and HFFs in chromosomes 21 and 22. Columns are different DEG windows, and rows are scoring methods; both are clustered using Ward's clustering with Euclidean distance. The purple scale values in heatmap cells represent the scores normalized across windows for each method. **b**, A heatmap of Pearson

correlation values for all pairs of scores on regions in **a**. **c**, The results of PCA of scores in **a**. **d**, A heatmap of Pearson correlation values for scores on predicted contact maps, generated through a series of perturbations (Extended Data Fig. 5a), for all pairs of the 13 methods that can be applied to predicted maps. The colors across the top of the heatmap identify the methods. Maps that resulted in no scores for any method were removed from the analysis.

the differences in how the raw Correlation and MSE prioritize map differences (Fig. 2). Furthermore, methods can be categorized into two main groups: one including the methods that correlate with each other, and the other including the ones that don't correlate well with the rest, namely TADcompare, Selfish, and HiC1Dmetrics ISC (Fig. 3b). This suggests that these methods provide somewhat different information about map changes as compared with the other techniques. On the second principal component, HiC1Dmetrics deltaDLR, SSC, and CIC cluster together, separate from the rest (Fig. 3c). Importantly, our four newly adapted methods—Eigenvector, Contact Directionality, Distance Enrichment, and Triangle—tend to score maps comparably to existing methods. These analyses give us a broad idea of which methods are concordant across a range of scenarios.

To investigate the unique capabilities of each method, we analyzed the top-scoring maps for each approach, that is, maps ranked as most different between HFFs and ESCs (Supplementary Fig. 3). We identified all maps that scored in the top 5% by only one method (Supplementary Fig. 4). We found that 14 of the 25 methods uniquely identify divergent maps (Extended Data Fig. 3), including our four newly adapted methods, highlighting the complementary information they provide. Some methods have high overlap among their top 5% scoring maps (Extended Data Fig. 4). For example, SCC and Correlation share 88% of their top maps, and neither of these methods has uniquely high-scoring maps (Extended Data Figs. 3 and 4). However, some methods are outliers, suggesting that they prioritize different features. For example, only 2% of the top maps ranked by MSE and Arrowhead are shared with respect to MSE top maps. Notably, methods based on loops—Loops, HiCcompare, and Selfish—do not cluster together or correlate well (Fig. 3a–c) and share 10% or less of their top maps (Extended Data Fig. 4). However, methods that focus on TADs—Arrowhead, TADs, and TADcompare—cluster relatively close together, especially Arrowhead and TADs (Fig. 3a,b), although they share only 8% or less of their top maps (Extended Data Fig. 4). These analyses highlight which methods are redundant and which are complementary.

The previous analyses focused on Micro-C data for 1-Mb windows at 2,048-bp resolution. To understand how different experimental techniques and analysis choices affect results, we compared methods using both Micro-C and Hi-C, at four resolutions (2,048, 4,096, 8,192, and 10,240 bp) and three window sizes (100 kb, 1 Mb, and 10 Mb). As expected, the scores for most methods agree well between Micro-C and Hi-C, although the two techniques do not match perfectly, given that Micro-C captures more detail (for example, local loops) (Supplementary Fig. 5)[43]. This agreement varies across window sizes: 1-Mb windows generally have higher agreement between the two methods than do 100-kb and 10-Mb windows (Supplementary Fig. 5). This is consistent with our finding that different methods cluster differently at varying window sizes (Supplementary Fig. 6). We hypothesize that, because different window sizes include different scales of genome structures (only loops at 100 kb, loops and TADs at 1 Mb, and compartments at 10 Mb), map ranking across methods changes on the basis of their individual focuses and biases. Nonetheless, certain methods cluster together regardless of window size, such as MSE, Insulation (mse), and Triangle (mse). Others, such as CHESS, are more affected by window size. However, most methods are robust to the choice of resolution, although there are exceptions, such as TADcompare and Loops (Supplementary Fig. 7). We also ensured that the data processing steps taken did not bias results (Supplementary Fig. 8). Altogether, these sensitivity analyses underscore the importance of window size in selecting a map comparison method and highlight methods that are sensitive to resolution and chromatin capture technology.

**Evaluation of comparison methods using in silico perturbation**
Above, we focused on experimentally determined contact maps derived from two human cell lines. Although these data contain many differences, they likely do not reflect the full range of biologically relevant contact map patterns. To expand our evaluation, we used machine-learning models that can accurately predict chromatin contact maps from DNA sequence alone[20,21,44]. Using predicted maps to benchmark map comparison methods, we can generate a huge number of map pairs with a variety of changes and patterns, overcoming the limitation that experimental contact maps are often highly conserved across cell types[7,39,45].

To generate map pairs with a wide range of differences, we perturbed sequences across the human reference genome in silico, generated contact maps for the perturbed and unperturbed sequence pairs using the Akita convolutional neural network model[20], and evaluated how each map comparison method scored every pair of maps (Extended Data Fig. 5a). We designed three types of perturbations: CTCF canonical motif insertions[46], endogenous CTCF motif deletions, and random 100-bp deletions (Methods), which in total produced 22,500 unique contact frequency map pairs. We used the same transformation and normalization process on the resulting scores as in the HFF versus ESC comparisons (Methods and Extended Data Fig. 6). Twelve of the existing methods cannot be applied to predicted maps because they cannot take locus-specific contact matrices as input and were therefore excluded from the analysis, resulting in 13 remaining methods (Supplementary Table 1). These include the four newly adapted methods, which can be applied to any distance-normalized map pairs, including predicted maps, highlighting one of their advantages.

We began with a qualitative evaluation, applying the 13 methods to map pairs representing a range of effect sizes. This confirmed that all methods are sensitive to large changes and insensitive to small ones (Extended Data Fig. 7). We then evaluated the top three highest-scoring map pairs for each method and found that, although all methods pick up maps that are different, MSE-based methods—MSE, Insulation (mse), and Triangle (mse)—pick out maps that have overall higher contrast (Supplementary Fig. 9).

We used PCA (Extended Data Fig. 5b) and correlation (Fig. 3d) to quantitatively assess similarities and differences between methods across the full set of 22,500 map pairs. These analyses revealed several trends. First, the TAD and Loop count-based methods have the lowest correlations with the rest of the methods (TADs and Loops mean correlation is $5.55 \times 10^{-17}$ and the rest of the methods mean correlation is 0.22, Fig. 3d). We hypothesize that this is the result of the small range of discrete scores generated by these methods, as well as their focus on a specific type of map change. We also found that MSE-based methods (MSE, Triangle_MSE, and Insulation_MSE) cluster separately from the other methods along the first principal component (Extended Data Fig. 5b). Clustering of the methods on the basis of their score vectors shows a similar trend (Extended Data Fig. 5c). This result aligns with our initial observation that Correlation and MSE often do not agree, especially across their top-scoring variants (Fig. 2, Extended Data Fig. 8, and Supplementary Fig. 10). Thus, the relationships between scores on predicted map pairs based on a range of sequence perturbations support the main conclusions from the experimental map comparisons.

**Simulations quantify method sensitivities and performance**
Sequence perturbations can produce a diversity of structural alterations, which often affect multiple aspects of a contact map at the same time. For instance, insertion of a CTCF site can both create a new TAD boundary and alter overall contact intensity. To disentangle how each method responds to changes, in particular map features, we generated simulated maps and synthetically altered one aspect at a time. We then measured the sensitivity of each method to each type of map change. As a template, we created a contact frequency map with two CTCF motifs forming a TAD ('Base' in Fig. 4) and used this canvas to simulate both biologically meaningful changes (for example, changes in TAD size, substructure, or intensity) and technical artifacts (for example, changes in noise or resolution) (Methods). For each change, we gradually increased the strength of the perturbation across 100 maps and

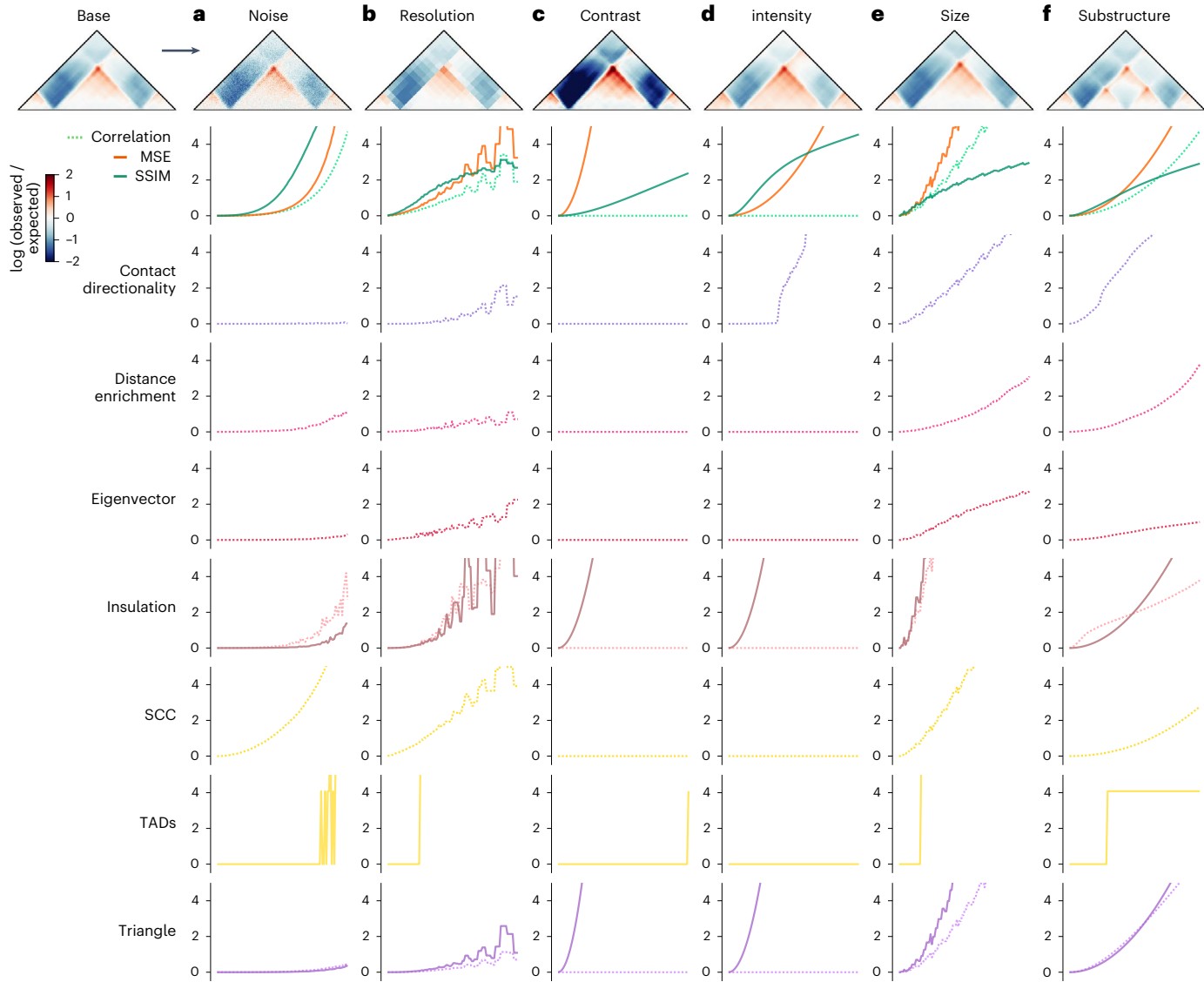

**Fig. 4 | Simulated contact frequency maps with controlled perturbations estimate disruption score method sensitivities. a–f,** Normalized disruption scores are plotted for a simulated contact frequency map containing a TAD across six types of perturbation. Each perturbation was added at 100 different degrees. The images shown correspond to the final degree—the maximum perturbation added. The line plots show the disruption scores obtained by comparing the original map (top left corner) with each perturbed map. The maps, corresponding to the incremental increases in perturbation, are shown alongside the change scores in Extended Data Figure 9. **a,** Noise is added by introducing random values drawn from a Gaussian distribution to the maps. **b,** Resolution is lowered by increasing bin size by averaging values in adjacent bins. **c,** Contrast is applied by increasing the range of the signal. **d,** Intensity is increased globally by adding a constant to all values. **e,** Size is increased by slightly enlarging the domain width. **f,** A sub-structure is added by gradually incorporating a new boundary at the center of the existing TAD. Dashed lines represent the correlation version of the method and solid lines the MSE version. Loops were excluded from the analysis since there are no loops intended to be created in these plots.

subsequently quantified the sensitivity of each method to each type of map change (Extended Data Fig. 9).

The methods responded differently across the simulated changes (Fig. 4). Global methods are most sensitive to technical variations, such as increased noise and decreased resolution, whereas contact map methods are more robust (Fig. 4a,b). The global methods have steeper curves, indicative of greater sensitivity to smaller perturbations, in contrast to the flatter curves of contact map methods. As expected, correlation-based methods are unaffected by changes in contrast and intensity, whereas MSE-based methods are highly sensitive (Fig. 4c,d). All methods except Eigenvector reliably identify TAD size and sub-structure changes. However, some prioritize certain types of organizational change (Fig. 4e,f). For example, Insulation and Triangle are sensitive to boundary changes, whereas Contact Directionality

highlights new boundaries but is less effective in identifying changes to existing boundaries. Eigenvector and Distance Enrichment are the least sensitive to changes likely to be technical, such as noise, resolution, contrast, and intensity. Furthermore, Triangle, Eigenvector, and Contact Directionality show the lowest sensitivity to increased noise and decreased resolution, compared with MSE, Correlation, and Insulation, highlighting the strengths of these methods.

We next used the simulated map framework to quantitatively evaluate the performance of each method. We generated a set of 'positive' map pairs with structural changes (changes in boundary) and a set of 'negative' map pairs with artificial changes (decreased resolution) (Fig. 5a). Although the positive map pairs don't encompass all possible changes and might favor methods that detect TAD or triangle-shaped changes, they provide a robust way to measure performance uniformly

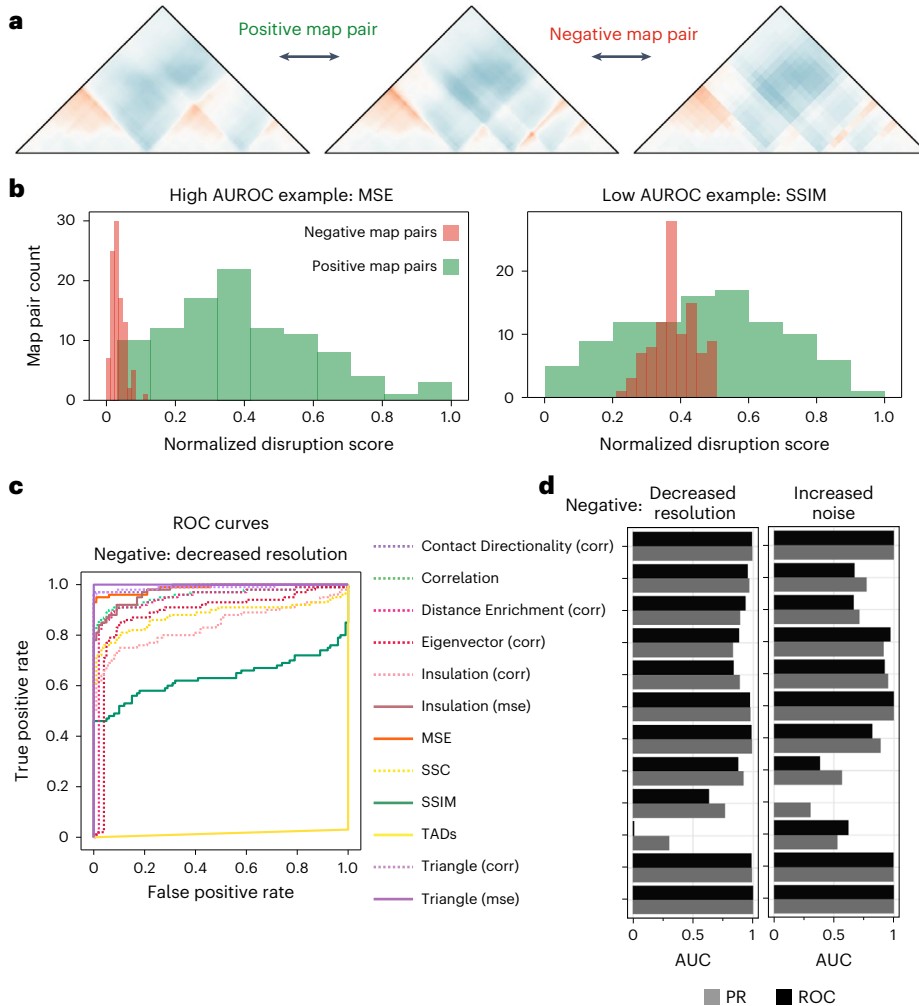

**Fig. 5 | Quantitative evaluation of contact map comparison methods.**
**a**, Examples of 'positive' and 'negative' map pairs reflecting a biologically meaningful change. The center map corresponds to a sequence with three CTCF-rich sites randomly added in the central 60% of the sequence, resulting in a structure with TADs, created using Akita. The left map (positive) was generated by removing the middle CTCF-rich site, which results in a changed structure; that is, the middle boundary is removed. The right map (negative) was generated by adding Gaussian noise to the center map, which results in the same structure but more noisy signal (Methods). **b**, Example distributions of a set of 100 positive and 100 negative (increased noise) map pairs and their normalized scores from MSE (high AUROC, 0.822) and SSIM (low AUROC, 0.000). **c**, ROC curves for 12 methods, each evaluated on sets of 100 positive and 100 negative (lower resolution) map pairs. **d**, The AUC values for the precision-recall (PR, gray) and ROC (black) curves across 12 methods. The negative set was either generated by adding noise to the maps (left bar plot) or by decreasing the resolution (right bar plot), both at random amounts for each map pair (Methods).

across methods. We calculated disruption scores and evaluated their ability to distinguish positives from negatives on the basis of the area under the curve (AUC) for the receiver operating characteristic (ROC) and precision recall (PR) curves. For most methods, there was a clear differentiation between scores from 'positive' and 'negative' map pairs, while for others there was not (Fig. 5b). This was reflected in the ROC and PR curves (Fig. 5c and Extended Data Fig. 10). MSE, Triangle (mse and corr), Contact Directionality (corr), and Insulation (corr) performed particularly well, further supporting the utility of our newly adapted methods. Others, such as SSIM, did not perform as well (Fig. 5c,d). To check the robustness of these trends to our definition of the negative set, we repeated the analysis with a 'negative' set generated by adding noise, which resulted in similar AUC values (Fig. 5d). Overall, PR curves exhibited trends that mirrored those observed with ROC analysis (Fig. 5c and Extended Data Fig. 10). Combining these findings enabled us to generate a scorecard for the evaluated methods (Table 1).

### Guidelines

Our study assessed 25 methods for comparing 3D genome contact maps (Supplementary Table 1). Through qualitative assessments, controlled sensitivity analyses, and performance evaluation, we determined which methods are similar or complementary across experimental techniques, genomic window sizes, and map resolutions. We found that most methods have unique strengths reflecting different sensitivities to biological and technical variation.

Global methods have many benefits. They are fast and easy to implement (Supplementary Table 1), require no predetermined decision-making about which features are important, and are consistent because they do not have tunable parameters. Nonetheless, they might have predispositions to prioritize certain patterns that might or might not have biological meaning. Correlation-based methods are insensitive to changes in contrast and intensity, whereas MSE-based methods are highly sensitive to them (Figs. 1 and 4). Although differences in contrast and intensity could be consequences of technical variability, they could also be biologically meaningful. Furthermore, global methods, especially MSE and SSIM, are particularly sensitive to increased noise and decreased resolution (Figs. 4 and 5).

In contrast to global methods, contact map methods can isolate specific changes of interest, such as changes to TAD boundaries or loops (Fig. 4 and Extended Data Figs. 2 and 9). At the same time,

**Table 1 | Contact map comparison methods differ in their performance and sensitivities**

| Comparison method | Performance | Sensitivities | | | |
|---|---|---|---|---|---|
| | High AUROC[a] (>0.75) | Resistant to noise | Resistant to decreased resolution | Indifferent to contrast change | Sensitive to TAD substructure changes |
| Correlation | ✓✓ | | ✓ | ✓✓ | ✓ |
| MSE | ✓✓ | | | | ✓✓ |
| SSIM | | | | ✓ | ✓ |
| Contact Directionality (corr) | ✓✓ | ✓✓ | ✓ | ✓✓ | ✓✓ |
| Distance Enrichment (corr) | ✓ | ✓✓ | ✓✓ | ✓✓ | ✓ |
| Eigenvector (corr) | ✓ | ✓✓ | ✓ | ✓✓ | |
| Insulation (corr) | ✓ | ✓ | | ✓✓ | ✓ |
| Insulation (mse) | ✓✓ | ✓✓ | | | ✓✓ |
| SCC | ✓ | | | ✓✓ | |
| Triangle (corr) | ✓✓ | ✓✓ | ✓✓ | ✓✓ | ✓✓ |
| Triangle (mse) | ✓✓ | ✓✓ | ✓ | | ✓✓ |
| TADs | | | | ✓ | ✓✓ |

Trends and patterns across disruption scores summarized from simulation analyses (Figs. 4 and 5). These features are evaluated qualitatively, taking into account various analyses and subjectively categorizing the results into three outcomes: no presence (blank), presence (single checkmark), and strong presence (double checkmark). No presence means that the method does not have a given feature, presence means that the method has the given feature, and strong presence means that the feature is more prevalent for that method. [a]High AUROC analysis is based on negative maps with decreased resolution.

they require more careful consideration. Contact map methods often require parameter tuning and significance thresholds, and are computationally and time intensive (Supplementary Table 1). These challenges can be addressed using default parameters and by decreasing resolution, although sensitivities to parameter tuning and resolution changes should be considered before a method is selected (Figs. 4 and 5 and Supplementary Figs. 1, 2, and 7). Finally, caution should be exercised when using TADs and Loops at scale, especially in maps without strong TADs or loops, where they can produce misleading results. Moreover, because they are count-based, these methods can rank map pairs only into groups of various count differences.

In general, the new and adapted methods we proposed align well with existing techniques, especially when comparing the top 5% of genome-wide scores—up to 86% are shared (Extended Data Figs. 4 and 8). Additionally, they are less sensitive to technical changes (Fig. 4), and can very accurately distinguish changes likely to have biological relevance from those with only technical differences (Fig. 5). Triangle stands out among the newly implemented methods, demonstrating high concordance with other methods, both for experimental and in silico maps (Extended Data Figs. 4 and 8, respectively), but it is also the slowest one (Supplementary Table 1). The Loops method also identifies most of the top 5% map pairs called by other methods for in silico maps (Extended Data Fig. 8), and it can identify uniquely top-scoring experimental maps (Extended Data Fig. 3).

Overall, we find that there is no 'one size fits all' metric that best identifies changes to every feature of interest in a chromatin contact map. Researchers should consider the intended application and the types of change that are meaningful when selecting the most effective and relevant metrics. In general, we recommend first applying global methods as an initial screen to identify the most disrupted maps, especially when evaluating large datasets. Using both correlation- and MSE-based scores will help mitigate weaknesses of each. We then suggest applying appropriate contact map methods to a subset of high-scoring map pairs to gain insight into the types of changes present. Finally, we recommend using methods that enable plotting of intermediate results, such as Eigenvector, Insulation, and Directionality index, for more qualitative analyses, such as visualizing changes and developing mechanistic hypotheses (Extended Data Fig. 2).

## Discussion

We evaluated and compared 25 methods for quantifying differences between pairs of 3D contact maps, including many methods that have not been previously used for this application. We introduced Eigenvector, Contact Directionality, Distance Enrichment, and Triangle. We found that the choice of scoring method can have a significant impact on the conclusions. Therefore, we suggest that multiple comparison metrics be used when seeking biological insights into the function of the 3D genome and provide guidance on which methods to use.

Several limitations should be considered when evaluating our results. Although we consider a range of experimental, predicted, and simulated maps, our findings might not apply to other experimental conditions, such as single-cell contact matrices or scenarios in which maps have a high level of noise and/or sparsity. Additionally, some of the methods we evaluated have parameters that can be tuned to optimize performance (Supplementary Table 1 and Supplementary Note). We did not set out to thoroughly evaluate all possible parameterizations, but rather focused on either default parameters or, as for TADs and Loops, selected a representative set on the basis of inspection of a few examples (Supplementary Fig. 1). Thus, methods with tuning parameters have potential to perform better than reported on specific applications. We also only tested three TAD callers—TADs, Arrowhead, and TADcompare—and three loop callers—Loops, Selfish and HiCcompare—to examine their general utility[41,42]. These represent methods in common use, but other implementations of these general approaches exist. We did not directly address the problem of identifying the threshold at which differences should be considered biologically or statistically significant. One could apply previously proposed[6,27,47] thresholding methods to the ranks computed with scoring methods to define a significant set of map pairs. Nonetheless, our evaluations quantified performance across the full range of scores for each method. Furthermore, although predicted maps provide an opportunity to evaluate large sets of map pairs with a variety of perturbations, they also have limitations, including the window size based on the input sequence of the model, a lack of compatibility with scoring methods that require .cool files (genome wide), and the inability to accept matrices (local) as input. Finally, differences in the input data and run times across methods, such as the use of whole-chromosome

data by some and locus-specific data by others, prohibited us from performing chromosome-wise comparisons.

We focused on quantitative methods for measuring differences between contact maps. Researchers often integrate additional data types to evaluate regions of interest. For example, to identify differences in genome structure between two cell types, one might overlay functional genomic annotations such as chromatin immunoprecipitation sequencing, assay for transposase-accessible chromatin sequencing or RNA sequencing, to identify regions in which the contact map difference aligns with a functional change. Developing and comprehensively evaluating methods that integrate functional genomic and structural data is a promising avenue for future work.

Overall, our study provides a foundation and framework for analyzing contact maps at scale. We provide practical and useful guidelines for scoring contact maps that will enable further discovery of the mechanisms of the 3D genome. Our codebase of methods enables flexible and fast scoring across contact maps in a unified framework. The experiments we performed as a part of this study, such as the in silico deletion and insertion of thousands of CTCF motifs, also provide a useful benchmark for evaluating diverse biological questions. Although we could not evaluate all possible methods and applications, we provide qualitative guidelines for users to make informed decisions when selecting a comparison method based on the scale and application of their research question. We anticipate that incorporating methods with greater biological interpretability, like those evaluated here, will also further improve machine-learning methods for predicting contact maps. Finally, several of the methods we investigated can be applied to other types of data, such as chromatin imaging, so our findings can likely inform genomic matrix comparisons beyond chromatin contact maps.

## Online content

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

## Methods

### Datasets

**Experimental maps.** Maps of 3D chromatin contacts are represented as 2D matrices of pairwise interaction frequencies. Regions with high values indicate genomic loci with a high frequency of interaction in physical space, on average. The experimental maps used in this study from HFFs and ESCs were preprocessed as training datasets for the Akita model[11,20], reflecting log (observed / expected) contact frequencies[48]. These high-quality Micro-C and Hi-C datasets underwent several normalization and processing steps, including normalization using genome-wide iterative correction (ICE)[30], adaptive coarse-graining, normalization for the distance-dependent decrease in contact frequency, log clipping to (−2,2), linearly interpolation to fill missing bins, and convolving with a 2D Gaussian filter for smoothing. The combination of observed-over-expected normalization with ICE enables removal of biases and the genomic-distance-dependent decay within the sample and scaling of the sequencing depth between samples[49]. Subsequent processing steps concentrate on locus-specific patterns, mitigate the impact of sparsity, and retain consistency across the experimental data and computational predictions. Our scoring metrics are minimally impacted by these steps, except for the log transformation, which leads to more sensitive detection of differences between map pairs (Supplementary Fig. 8).

The genomic regions that we used for comparison were either genome-wide tiled regions (Fig. 2), select examples (Fig. 3), or regions surrounding DEGs between ESCs and HFFs (Fig. 4). The genome-wide tilted regions were obtained by sliding across genomic contigs, which were generated by splitting the genome at assembly gaps, large unmappable regions, and low-coverage regions. Each tile had a window size of $1 \times 2^{20}$ bp and stride of $1 \times 2^{18}$ bp (~262 kb) or $1 \times 2^{19}$ bp (~524 kb)[20], to generate 7,840 windows of 1 Mb. To identify DEGs, we used edgeR with RNA-sequencing data downloaded from the 4DN data portal (https://data.4dnucleome.org/). The windows around the DEGs were the 100-kb, 1-Mb, or 10-Mb regions around the gene, centering the gene body in the window. When the region fell outside of chromosome-arm coordinates, we removed it from the analysis. The comparison of DEG-adjacent regions was done only for chromosomes 21 and 22—the two smallest chromosomes—because some of the methods, such as TADcompare and HiCcompare, are too computationally intensive to run on larger chromosomes without specialized computing resources. We then removed any windows in which more than 60% of the map had a value of 0, indicating potential data unreliability. This threshold was set by looking at maps at different thresholds and determining where they began to look artifact-based. As a result, we identified 201 10-kb regions, 256 1-Mb regions, and 285 10-Mb regions across the two chromosomes (Supplementary Table 2). For the 10-kb and 1-Mb windows, we used 2,048-bp bins—the same resolution as Akita—to maintain consistency. For 10-Mb windows, we used 20,480-bp bins so that the maps contained a similar number of bins across window sizes.

**In silico perturbed maps.** To facilitate large-scale comparisons of contact maps, we generated thousands of maps predicted from in silico CTCF-motif insertions, CTCF-motif deletions, and deletions of random 100-bp sequences. These alterations were passed into Akita[20], which predicts genome folding from sequence, enabling the creation of pairs of maps with structural rearrangements. For CTCF insertions, CTCF motif sequences were randomly selected from annotated CTCF sites in the reference genome from the hg38 build of the JASPAR database[46]. These motifs were inserted into the center of 1 Mb of DNA with start locations randomly selected from chromosome 1. Akita requires a fixed input of $1 \times 2^{20}$ bp. Additional sequence was trimmed from the 3′ end, such that the final sequence remained 1 Mb. To curate deletions, we again selected random CTCF sites from JASPAR, pulled the surrounding 1 Mb of DNA, removed the motif sequence, and pulled in additional sequence from the 3′ end such that the entire sequence remained 1 Mb in length. The same strategy was applied to randomly selected 100-bp

fragments for deletion. All generated 1-Mb genomic query sequences were filtered to exclude overlap with ENCODE blacklisted regions[50]. For each perturbation, Akita was provided with both the original genomic sequence and the perturbed sequence, resulting in two predicted contact maps, each 448 × 448 pixels in size. Each pixel has a resolution of 2,048 bp, representing the center 917,504 bp of a total length of ~1 Mb of DNA sequence[20]. The dataset consists of 7,500 contact map pairs for each category of perturbation, totaling 22,500 pairs.

**Simulated maps.** To generate simulated maps, we initially generated predicted maps with Akita using random DNA sequences. The predicted maps showed minimal structure. To eliminate any higher-order folding patterns, we shuffled sequence matches to the forward and reverse canonical CTCF motifs[46] to produce a predicted blank canvas map. Structure was reintroduced to simulated maps by inserting forward and reverse CTCF motifs one-fourth and three-fourths through the random DNA sequence, producing TAD-like boundaries.

To evaluate method sensitivity (Fig. 4), we tuned simulated parameters as described below. Visualizations of these changes are available in Extended Data Figure 9.

- Noise: Gaussian noise was added to the maps with a s.d. ranging from 0 (no added noise) to 0.2.
- Resolution: the original 448 × 448 map was downsampled ranging from a resolution of 2,048 bp (original resolution) to 50,972 bp, by averaging neighboring bins into larger bins.
- Contrast: pixel intensities of the contact map were multiplied by a scalar ranging from 1 (no increase in contrast) to 2.
- Intensity: a scalar value ranging from 0 (no addition) to 0.2 was added to all pixels in the contact map.
- Size: the size of the substructure within the map was increased by resizing the original map by a scalar and trimming the matrix back down to the original dimensions. Map sizes were increased by a factor of 1 (no resize) to 1.1.
- Substructure: an additional map was created by introducing CTCF halfway into the random sequence to produce an additional boundary. The original map was combined with the substructure map, with a multiplier ranging from 0 (no added structure) to 1 (total added structure).

Map differences were scored between the base map and each map with a tuned parameter. Scores were adjusted, when necessary, so that larger values always correspond with bigger differences. Then, scores were scaled on the basis of 100 random deletions chosen randomly from a total of 7,500 (Extended Data Fig. 5), with each score divided by the mean of the 100 scaling scores. This process enables comparison of scores across methods (Fig. 4).

To measure method performance, we generated positive and negative map pairs based on a neutral map containing a TAD. Positive maps were generated by perturbing the sequence to alter the structure, and negative maps were produced by degrading the resolution or increasing noise. Neutral maps began with a blank canvas map (no structure), as described above, and were formed by inserting a sequence of three consecutive CTCF motifs at three random locations in the sequence between the 20th and 80th percentile of the sequence length. The left CTCF cluster featured reverse motifs, while the middle and right clusters contained forward motifs. To generate positive maps, the middle CTCF motif cluster was removed from the neutral sequences. Depending on the location of the middle removed CTCF cluster, the positive maps either lost a central boundary or experienced a reduction in boundary strength owing to the loss of a CTCF cluster adjacent to another. These manipulations resulted in 100 map pairs with a variety of structural differences. For negative maps, the neutral map was modified by introducing Gaussian noise with a s.d. randomly selected between 0.01 and 0.1, or by decreasing the resolution to a bin size randomly chosen between 25,00 to 25,000 bp. This resulted in two

sets of 'negative' map pairs, one based on noise and one on resolution. The positive and negative map pair sets were scored using each of the 12 applicable methods for predicted contact maps. The scores were min-max normalized for each method so that a score of 0 means there is no change (predicted negative) and a score of 1 means there is change (predicted positive). Using the true 'positive' and 'negative' labels of the map pairs, ROC and PR curves were generated, and the AUC was calculated for each method using sklearn (Fig. 5 and Extended Data Fig. 10).

### Comparing methods

**Adapted methods.** We describe four new or adapted methods—Contact Directionality, Distance Enrichment, Eigenvector, and Triangle—to compare contact frequency maps generated from experiments or predictive models. Contact Directionality, Distance Enrichment, and Eigenvector were adapted from Directionality Index[39], Contact Decay[9], and PCA[38], respectively, which are established methods for analyzing individual experimental maps at different scales. Triangle was developed to detect changes across gradual scales of contact, from regions that are near to far. Our goal was to evaluate maps on the basis of characteristics that these methods focus on. For example, Contact Directionality should evaluate changes at TAD boundaries where contact is concentrated in either direction but not at the center. Distance Enrichment is meant to emphasize changes at regions that are distal since it measures the average contact across all distances, over-representing distal regions that have fewer bins in the map. Eigenvector is meant to identify changes to blocks of regions that have more intra-region contact than inter-region contact. These methods were applied to each map, and the resulting tracks were compared using Correlation or MSE. The intermediate tracks can be visualized to qualitatively evaluate differences along the map (Extended Data Fig. 2). However, Triangle is applied directly to the map pair with no intermediate track. More detailed information on these methods and their implementation can be found in the Supplementary Note.

**Scoring contact maps.** We applied all comparison metrics to pairs of experimental, predicted, and synthetic maps. For details on computation of each metric, see the Supplementary Note. Missing values were masked before evaluation and were not considered by the comparison metrics. Implementations of scoring methods can be found in the codebase. MSE and/or Correlation were applied to some methods to collapse two 2D tracks into a scalar value. Pearson correlation behaved almost identically to Spearman's rank correlation and therefore was excluded from analyses. To expedite evaluation time across thousands of comparisons of in silico perturbed map pairs (Extended Data Figs. 5, 6, and 8), the resolution of the input was reduced by fivefold for Triangle, a process known to be computationally intensive. Furthermore, for Contact Directionality, we changed the resolution to 2,000 because some of the resulting tracks in some maps extended to infinity. Finally, for in silico perturbed maps, Loops and TADs scores were missing in cases where the feature was not detected in either map.

To ensure that scores were comparable across approaches, we adjusted some methods such that higher values indicate greater disruption and smaller values indicate that maps are more similar; for example, for methods such as Correlation, we used the formula 1 − Correlation to ensure that higher scores signify bigger differences between maps. For all the results, we use min-max normalized scores to make it easier to interpret how scores for one method compare to scores of another. For predicted maps, we additionally scale all values by the mean score of all random 100 bp deletions using Akita, which we find to have minimal impact (Extended Data Fig. 6). For example, a raw MSE of 0.0065 and a raw 1 - pearson correlation of 0.036 both correspond to the same normalized score of 2. That is, a disruption of that magnitude corresponds to 2 times the average disruption of a 100 bp deletion.

For Loops and TADs, we quantify the ratio of changed (for example, added or lost) features (TADs or loops) to extend these approaches and generate a single score for each pair of maps.

**Method parameters.** The following methods required no adjustable input parameters: MSE, Spearman's rank correlation coefficient, and Pearson correlation coefficient, SSIM, SCC, Distance Enrichment, Eigenvector, and Triangle correlation. We describe tunable parameters choices for the remaining methods below. Parameters were set to default unless otherwise noted. We did not optimize tunable parameter choices, instead selecting default choices from existing approaches. Results from alternative parameter selection are demonstrated in Supplementary Figures 1 and 2.

For Insulation, the window_size parameter was set to 10, meaning the size of the diamond-shaped window was 10 bins.

For Contact Directionality, the parameter 'window_resolution' was set to 10,000 bp, which determines the resolution of the sliding window. The 'replace_ends' option was invoked to replace the values at the ends of the directionality index track with zeros. The 'buffer' parameter was set to 50, meaning that zeros are applied within 50 base pairs from the track ends.

For Loops, the parameter p was set to 2, which defines the width of the interaction region surrounding the peak. The width was set to 5, determining the size for the donut filter. The ther parameter was set to 1.1, which specifies the threshold for the ratio of the center windows to both the donut and lower-left filters. Similarly, ther_H and ther_V were both set to 1.1, defining the threshold for the ratio of center windows to the horizontal and vertical filters, respectively. Finally, radius was set to 5, which determines the maximum distance between two loop points for them to be considered part of the same loop.

For TADs, the window_size was set to 5, defining the size of the diamond-shaped window. The ther parameter was set to 0.2, which established the threshold for TAD boundaries. Finally, the radius was set to 5, specifying the maximum distance between two TADs for them to be considered part of the same TAD.

### Data analysis

The following packages and versions were used: Python v3.10.12, cooler v0.9.2, cooltools v0.5.4, Matplotlib v3.7.2, Numpy v1.23.5, Pandas v1.5.3, scipy v1.10.1, Seaborn v0.12.2, h5py v3.8.0, hicrep v0.2.6, sklearn v1.0.2, skimage v0.19.3, Arrowhead of Juicer Tools v1.8.9, CHESS v0.3.8, HiC1D-metrics v0.2.5, Selfish v1.14.0, Rstudio v0.16.0, HiCcompare v1.26.0, TADcompare v1.14.0, dcHiC v1, and conda v4.12.0.

### Reporting summary

Further information on research design is available in the Nature Portfolio Reporting Summary linked to this article.

## Data availability

Micro-C and Hi-C datasets used in map comparisons and RNA-seq datasets for identifying DEG genes are publicly available from the 4DN data portal (Micro-C for ESCs and HFFs: 4DNES21D8SP8 and 4DNESWST3UBH; Hi-C for ESCs and HFFs: 4DNESX75DD7R and 4DNESNMAAN97; and RNA-seq for ESCs and HFFs: 4DNES3IOYG74 and 4DNESFH3EHTU). The reference genome from the hg38 build was used.

## Code availability

Our codebase is publicly available to enable researchers to easily test and apply all 25 methods. The code is written in Python and R and accompanied by documentation to help users get started. The methods enable flexible parameter setting and running multiple methods simultaneously on one dataset, making it easier to compare the results of different approaches and to select the most appropriate methods. To aid in interpretation, we also provide code for visualizing methods that generate intermediate 1D tracks as in Extended Data Figure 2a. Overall, our codebase provides a valuable resource for researchers who wish to apply multiple contact map comparison methods to their own datasets

and rank pairs of maps based on their differences. All original code and resulting data for experimental and in silico scored contact map pairs are available at https://github.com/pollardlab/contact_map_scoring and https://doi.org/10.5281/zenodo.13977146 (ref. 51).

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

## Acknowledgements

We gratefully acknowledge members of the Pollard and Capra labs for project feedback, as well as V. Ramani, for proposing that we include Eigenvector in the study. We additionally thank M. Keiser for feedback and support.

This work was supported by the NIH 4D Nucleome Project (award no. U01HL157989 to K.S.P.), the NIH Office of the Director (award no. R03OD034499 to K.S.P.), NIGMS (award no. R35GM127087 to J.A.C, award no. T32GM007347), NHGRI (award no. F30HG011200 to E.M.), Additional Ventures, two UCSF Achievement Rewards for College Scientists Scholarship (K.G. and L.M.G.) and Gladstone Institutes.

## Author contributions

Conceptualization: K.G., L.M.G., S.K., E.M., M.P., J.A.C., K.S.P. Methodology: K.G., L.M.G., S.K., E.M. Formal analysis and investigation: K.G., L.M.G., S.K., E.M. Writing, original draft: K.G., L.M.G., S.K., E.M., M.P. Writing, review and editing: K.G., L.M.G., S.K., E.M., M.P., J.A.C., K.S.P. Visualization, K.G., L.M.G., S.K., E.M. Supervision, J.A.C., K.S.P. Funding acquisition, J.A.C., K.S.P.

## Competing interests

The authors declare no competing interests.

## Additional information

**Extended data** is available for this paper at https://doi.org/10.1038/s41592-025-02630-5.

**Correspondence and requests for materials** should be addressed to John A. Capra or Katherine S. Pollard.

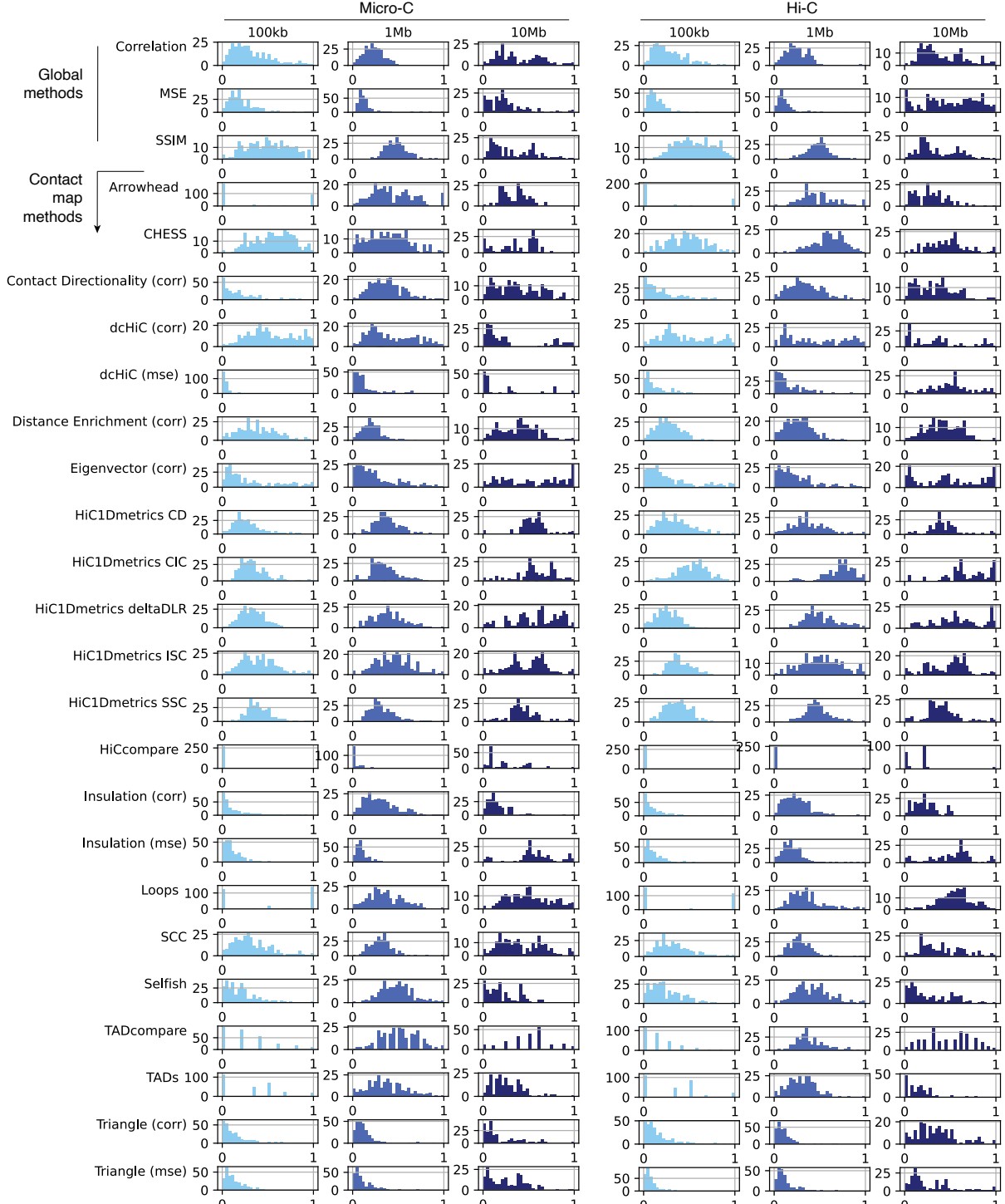

**Extended Data Fig. 1 | Score distributions of DEG regions for 10 kb, 1 MB, and 10 Mb windows.** Each disruption score method (rows) produces a different range and mean (red line) of distribution of scores. Histograms show the normalized scores comparing maps between ESCs and HFF around DEGs of window size 10 kb (left), 1 Mb (middle), and 10 Mb (right), for MicroC (left panel) and HiC (right panel).

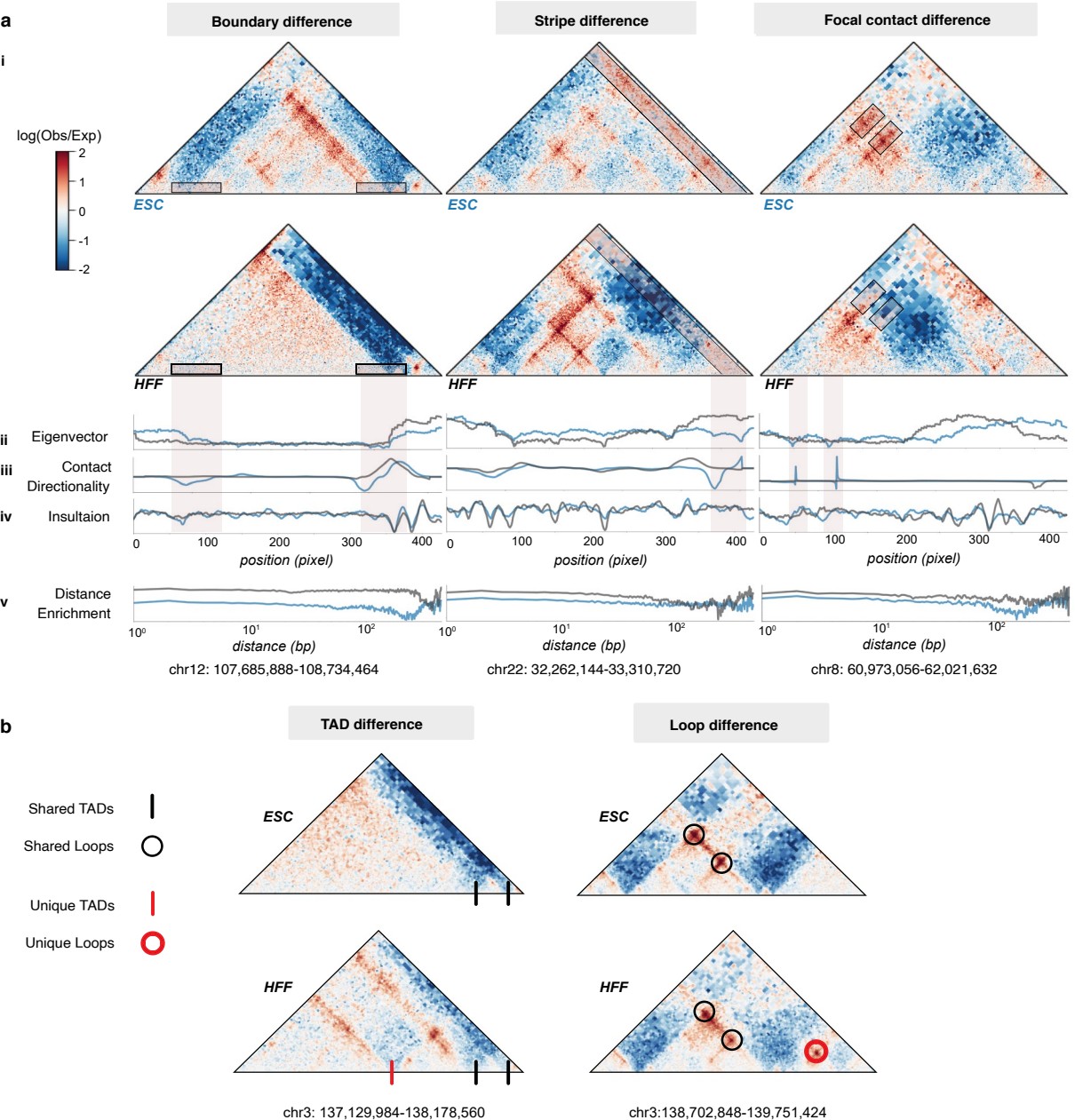

**Extended Data Fig. 2 | Visualizing changes that certain methods detect on example map pairs. a. i.** Examples of regions where contact frequency maps differ between HFF and ESCs MicroC across three structural changes: a lost TAD boundary (left panel), a lost stripe (middle panel), and lost loops (right panel), as marked by red boxes. ***ii-vi.*** Tracks of scores along the map for ESCs (blue) and HFF (gray) for applicable methods. Tracks for methods in (ii - v) correspond to the coordinates of the contact maps, while Distance enrichment (v) is plotted across genomic distance. Contact directionality compares the interaction tendency of each locus towards upstream or downstream regions between maps (see Supplementary Method Descriptions for detailed equation).

Distance enrichment is the comparison of the track of average interaction frequencies along genomic distances generated for each map. Disruption scores for these methods are calculated by taking the Spearman's correlation or MSE between the two tracks. These examples highlight different features in the tracks, from overall structural differences and average contact, to sharp changes in contrast. **b.** Two example loci with differences in TADs (left panel) or loops (right panel) between HFF and ESC, where the TADs are marked with lines and the loops are circled. Black boundaries and circles are shared between the maps while red boundary lines and circles are unique to one of the maps.

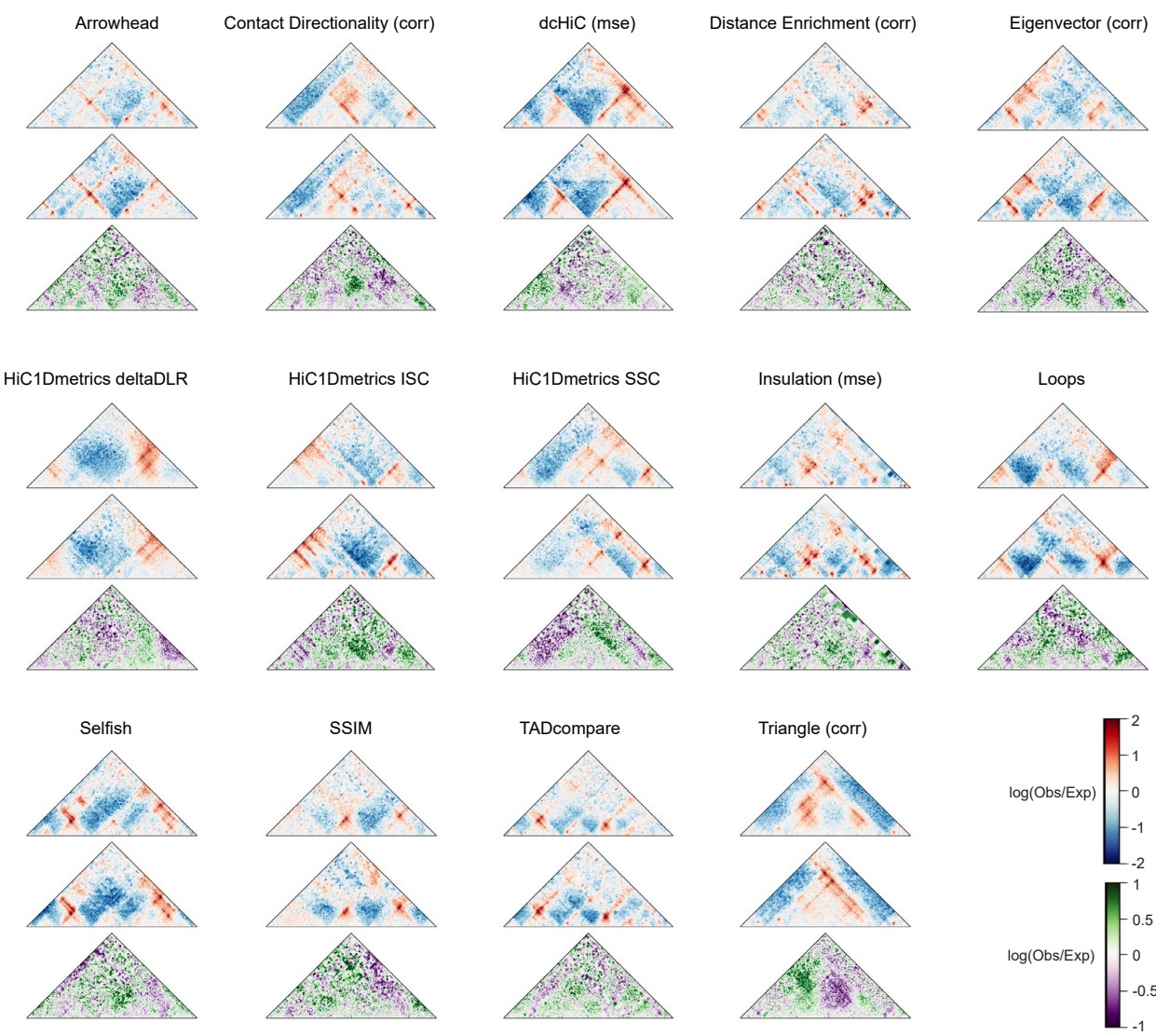

**Extended Data Fig. 3 | Uniquely high scoring map for each method.** DEG window scores for 1 Mb windows between MicroC ESC and HFF were scored and the top 5% highest scoring maps for each method were compared to get those that were top scoring in only one method. Here we show an example map out of all the uniquely top scoring maps for each method (total of 60 maps across 16 methods).

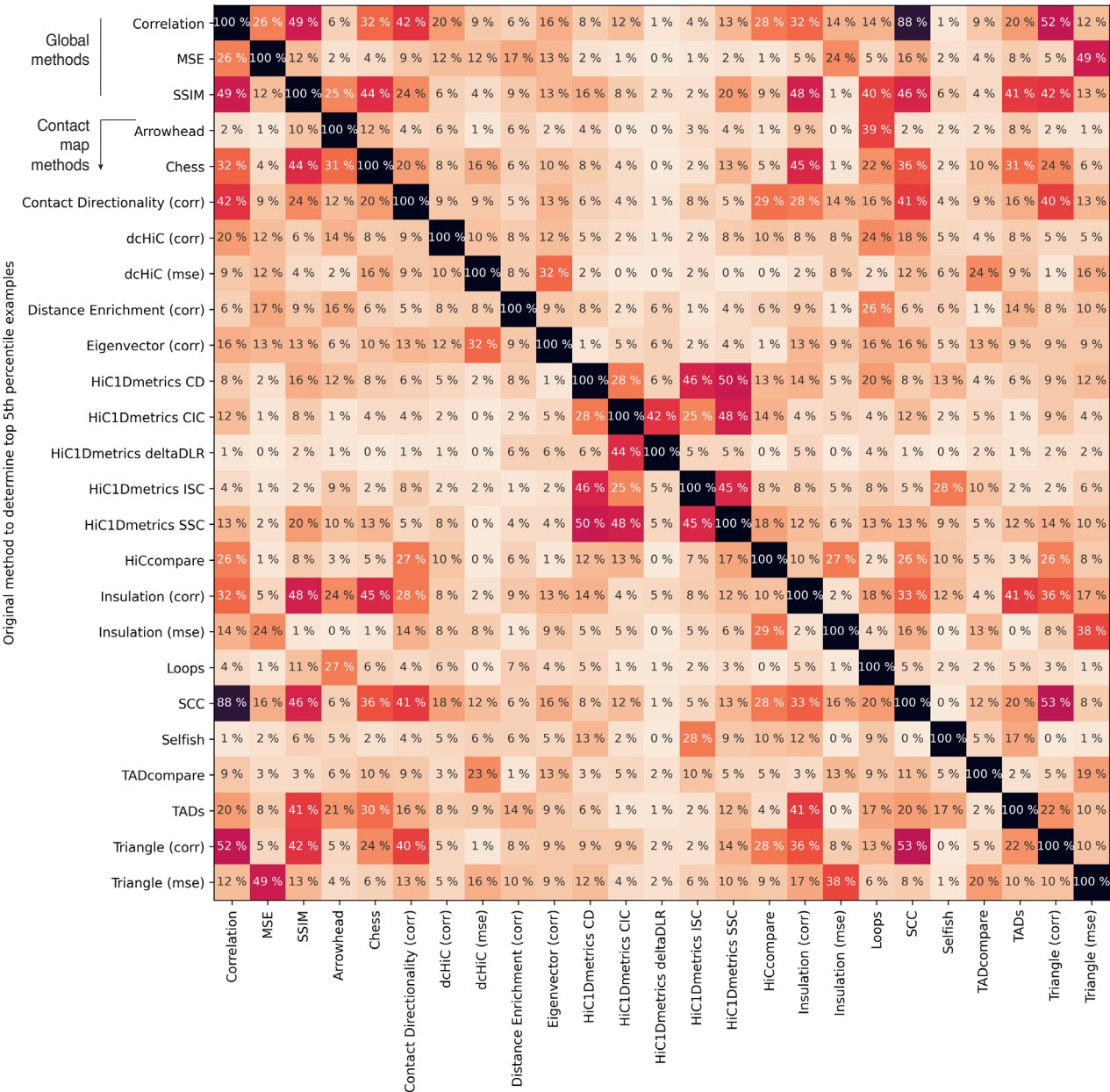

**Extended Data Fig. 4 | Overlap of the most disruptive windows identified by each scoring method.** MicroC ESC and HFF 1 Mb windows in chromosomes 21 and 22 were compared. Each cell in the heatmap represents the percentage of map pairs that are above the 5% cutoff for the method in row and above the 5% cutoff for the method in the column with respect to all map pairs that are above 5% cutoff in the method in the row. Darker colors indicate higher concordance for the top scoring loci. The heatmap is symmetric except for methods with a smaller range of values. The imbalance of these two methods is caused by multiple map pairs that have scores equal to the 5th percentile, which results from methods producing low counts of discrete values.

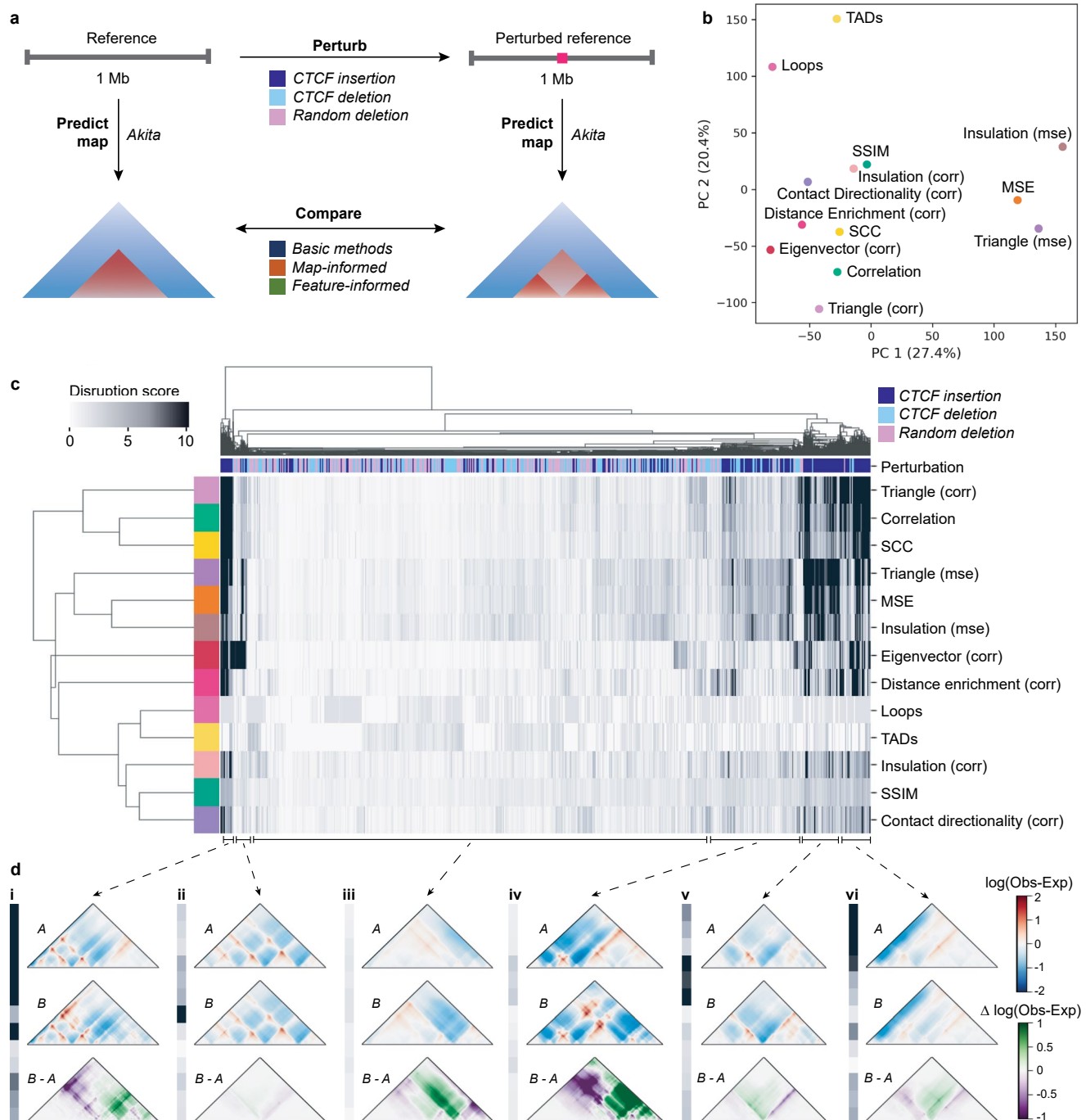

**Extended Data Fig. 5 | Comparison of disruption score methods. a**. Schematic describing the strategy for comparing in silico perturbed contact maps. Random -1 Mb windows of the human genome (GRCh38) are selected and input into Akita to predict chromatin contacts (left). The same window is also perturbed with a CTCF motif insertion, deletion, or random 100 base pair deletion. The resulting sequence is also input into Akita to predict chromatin contacts of this perturbed reference sequence (right). The perturbed and unperturbed maps were compared by applying the 13 global and contact map methods. **b**. Principal component analysis of disruption scores of each method from perturbed map pairs. **c**. Heatmap of normalized disruption scores across all methods and perturbations. The colored key along the top of the heatmap indicates whether

the perturbation was a random deletion (pink), a CTCF insertion (navy), or a CTCF deletion (light blue). Method colors are the same as in b. Six broad trends in disruption score patterns across methods are marked with brackets. **d**. Representative example map pairs chosen from the groups identified in c. Example maps were manually selected by visual inspection of the pattern of scores (grayscale heatmap) that most closely matched the pattern in each section of the heatmap: i. high scores across 5 methods; ii. low across all methods except for Eigenvector (corr); iii. low scores across all methods; iv. low scores across methods but higher for MSE-based scores; v. high scores only for MSE-based scores; vi. high scores for correlation-based scores: triangle (corr), corr, and SCC.

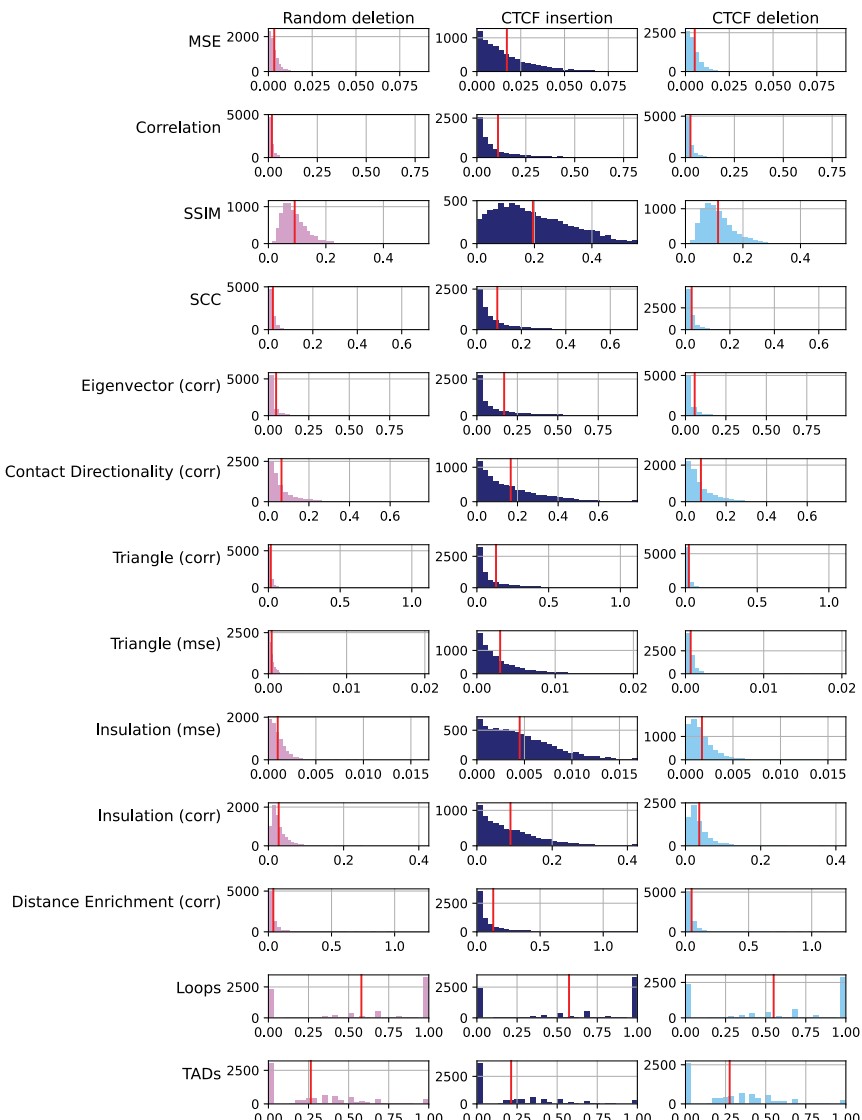

**Extended Data Fig. 6 | Score distributions of random deletions, CTCF deletions, and CTCF insertions.** Each disruption score method (rows) produces a different range and mean (red line) across scores produced. Histograms show the raw scores comparing maps produced by 7500 random 100 bp deletions (left), 7500 CTCF insertions (middle), and 7500 CTCF deletions (right). To enable comparisons between the different scores, the rest of the *in silico* perturbation map pair figures report scores standardized to the mean disruption produced by a random 100 bp deletion.

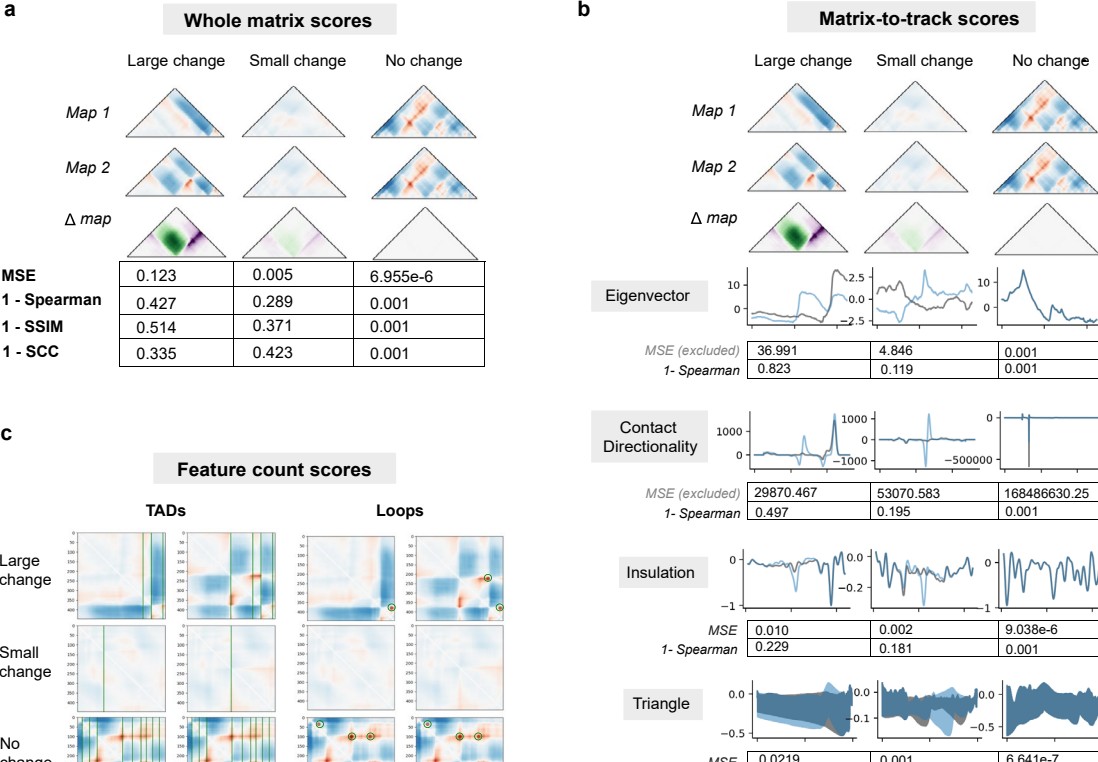

**Extended Data Fig. 7 | Scoring metrics on contact map pairs with large, small, and minimal changes. a**. Scoring results across three example loci with a large, small, and minimal change upon CTCF motif insertion for whole matrix methods that take in contact matrices. **b**. Scoring results across three example loci for Matrix-to-track methods that produce tracks for each map and generate scores by comparing those tracks with correlation (1- Spearman) or MSE. Raw tracks are shown for each measurement and the MSE and 1- Spearman's correlation between the tracks are shown below. **c**. Scoring examples across three example loci with no change, a minimal change, and a large change to folding for Feature-count methods that count features and generate scores by calculating the overlap ratio of features between the maps.

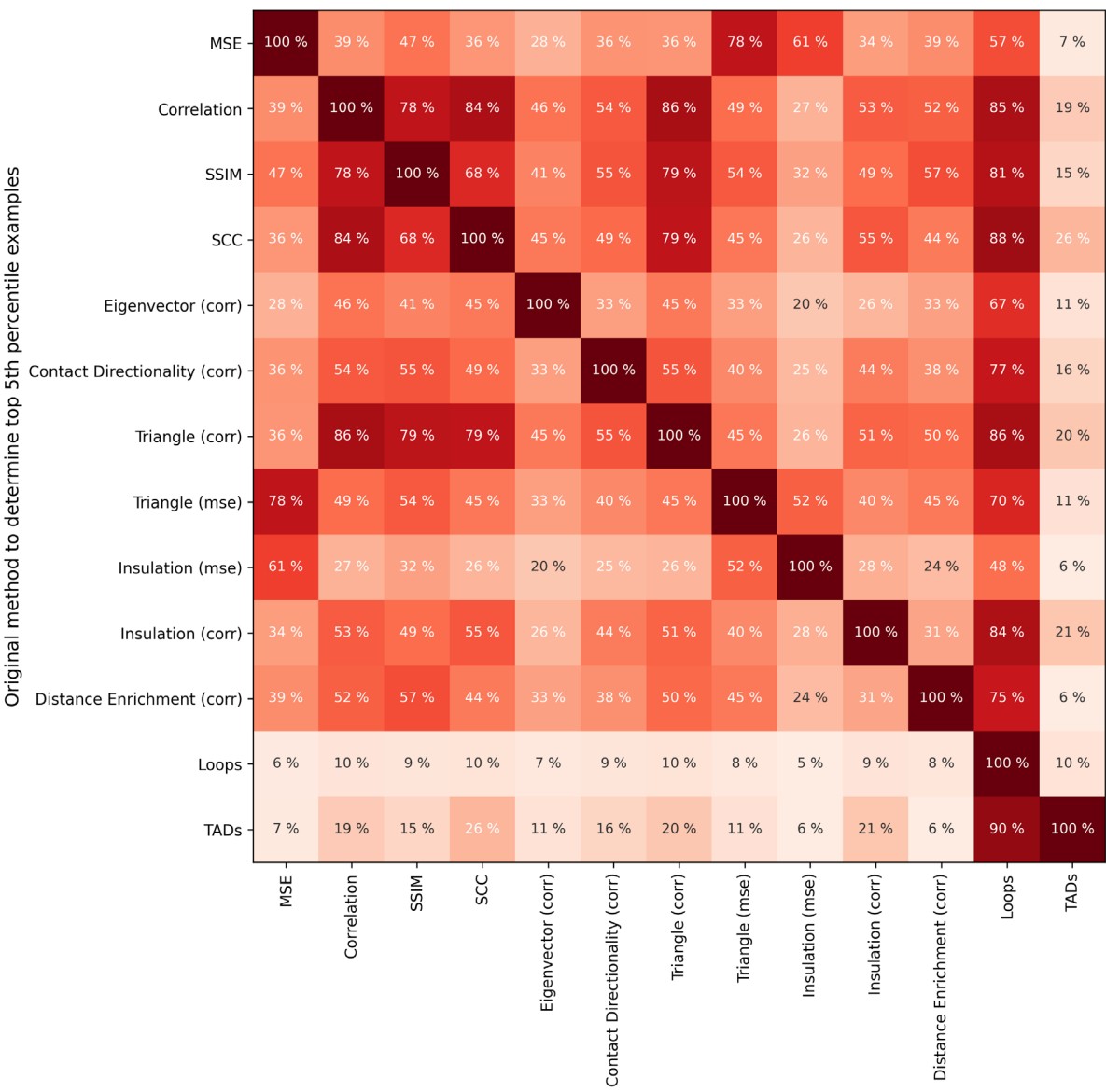

**Extended Data Fig. 8 | Overlap of the most disruptive map pairs identified by each scoring method.** Each cell in the heatmap represents the percentage of map pairs that are above the 5% cutoff for the method in row and above the 5% cutoff for the method in the column. Darker colors indicate higher concordance for the top scoring loci. The heatmap is symmetric except for Loops and TADs. The imbalance of these two methods is caused by multiple map pairs that have scores equal to the 5th percentile, which results from methods producing low counts of discrete values.

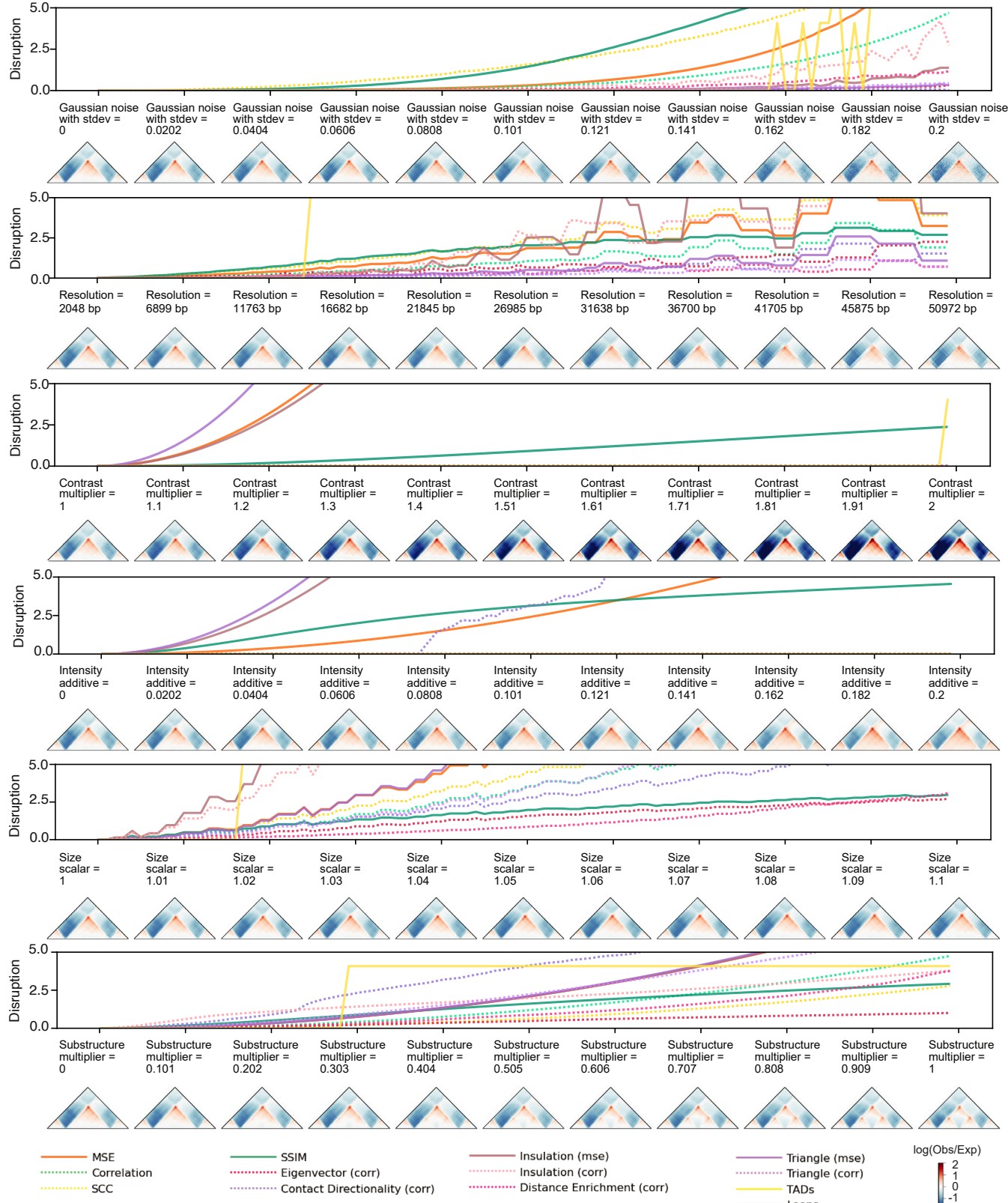

**Extended Data Fig. 9 | Changes of disruption scores with gradual increases in perturbations.** Each subpanel shows the changes of disruption scores (top row) and contact maps (bottom row) against the incremental changes in a technical or biological variation. The colors of the scoring metric are the same as seen in Figs. 4 and 5.

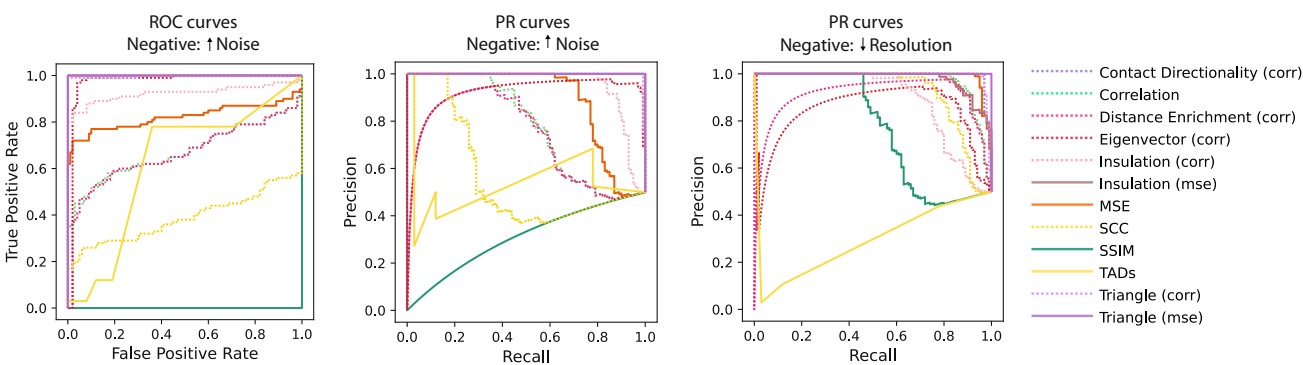

**Extended Data Fig. 10 | ROC and precision recall curves.** ROC (left) and precision recall curves using a negative set with noise (middle) or resolution (right) for all the methods that we applied to predicted maps.

|---|---|

# Reporting Summary

## Statistics

For all statistical analyses, confirm that the following items are present in the figure legend, table legend, main text, or Methods section.

| n/a | Confirmed | |
|---|---|---|
| ☐ | ☒ | The exact sample size (*n*) for each experimental group/condition, given as a discrete number and unit of measurement |
| ☒ | ☐ | A statement on whether measurements were taken from distinct samples or whether the same sample was measured repeatedly |
| ☐ | ☒ | The statistical test(s) used AND whether they are one- or two-sided *Only common tests should be described solely by name; describe more complex techniques in the Methods section.* |
| ☒ | ☐ | A description of all covariates tested |
| ☒ | ☐ | A description of any assumptions or corrections, such as tests of normality and adjustment for multiple comparisons |
| ☐ | ☒ | A full description of the statistical parameters including central tendency (e.g. means) or other basic estimates (e.g. regression coefficient) AND variation (e.g. standard deviation) or associated estimates of uncertainty (e.g. confidence intervals) |
| ☐ | ☒ | For null hypothesis testing, the test statistic (e.g. *F*, *t*, *r*) with confidence intervals, effect sizes, degrees of freedom and *P* value noted *Give P values as exact values whenever suitable.* |
| ☒ | ☐ | For Bayesian analysis, information on the choice of priors and Markov chain Monte Carlo settings |
| ☒ | ☐ | For hierarchical and complex designs, identification of the appropriate level for tests and full reporting of outcomes |
| ☐ | ☒ | Estimates of effect sizes (e.g. Cohen's *d*, Pearson's *r*), indicating how they were calculated |

*Our web collection on statistics for biologists contains articles on many of the points above.*

## Software and code

Policy information about availability of computer code

| Data collection | No software was used for data collection. |
|---|---|
| Data analysis | python packages: Python v3.10.12, cooler v0.9.2, cooltools v0.5.4, Matplotlib v3.7.2, Numpy v1.23.5, Pandas v1.5.3, scipy v1.10.1, Seaborn v0.12.2, h5py v3.8.0, hicrep v0.2.6, sklearn v1.0.2, skimage v0.19.3, Arrowhead of Juicer Tools v1.8.9, CHESS v0.3.8,  HiC1Dmetrics v0.2.5, Selfish v1.14.0. R packages: Rstudio v0.16.0, HiCcompare v1.26.0, TADcompare v1.14.0, dcHiC v1. conda v4.12.0. Other: Akita v1. All original code and resulting data for experimental and in silico scored contact map pairs are available at https://github.com/pollardlab/contact_map_scoring. |

For manuscripts utilizing custom algorithms or software that are central to the research but not yet described in published literature, software must be made available to editors and reviewers. We strongly encourage code deposition in a community repository (e.g. GitHub). See the Nature Portfolio guidelines for submitting code & software for further information.

## Data

Policy information about <u>availability of data</u>

All manuscripts must include a <u>data availability statement</u>. This statement should provide the following information, where applicable:
- Accession codes, unique identifiers, or web links for publicly available datasets
- A description of any restrictions on data availability
- For clinical datasets or third party data, please ensure that the statement adheres to our <u>policy</u>

Micro-C and Hi-C datasets used in map comparisons and RNA-seq datasets for identifying DEG genes are publicly available from 4DN data portal (Micro-C for ESC and HFF: 4DNES21D8SP8 and 4DNESWST3UBH, Hi-C for ESC and HFF: 4DNESX75DD7R and 4DNESNMAAN97, and RNA-seq for ESC and HFF: 4DNES3IOYG74 and 4DNESFH3EHTU). The reference genome from the hg38 build was used.

Resulting data for experimental and in silico scored contact map pairs have been deposited at https://github.com/pollardlab/contact_map_scoring.

## Human research participants

Policy information about <u>studies involving human research participants and Sex and Gender in Research.</u>

| | |
|---|---|
| Reporting on sex and gender | N/A |
| Population characteristics | N/A |
| Recruitment | N/A |
| Ethics oversight | N/A |

Note that full information on the approval of the study protocol must also be provided in the manuscript.

# Field-specific reporting

Please select the one below that is the best fit for your research. If you are not sure, read the appropriate sections before making your selection.

☒ Life sciences    ☐ Behavioural & social sciences    ☐ Ecological, evolutionary & environmental sciences

For a reference copy of the document with all sections, see nature.com/documents/nr-reporting-summary-flat.pdf

# Life sciences study design

All studies must disclose on these points even when the disclosure is negative.

| | |
|---|---|
| Sample size | Results are based on a sample size of 256 1-megabase regions of human chromosomes 21 and 22. We chose these two chromosomes since they are the smallest chromosomes, and some of the methods we evaluated have computational limitations and would not run on more data. After quality control filtering, this results in a sample size of 256 regions for comparison. |
| Data exclusions | Contact maps that were missing at least 60% of the information were excluded from the study. |
| Replication | Our benchmark results focus on differences between methods on the same input data. The benchmark includes data from the same genome region across biological replicates (different cell lines) and technical replicates (different Hi-C experiments). We did not include any other forms of replication experiments. |
| Randomization | There are no experimental groups in this study. |
| Blinding | There was no group allocation for this study. |

# Reporting for specific materials, systems and methods

We require information from authors about some types of materials, experimental systems and methods used in many studies. Here, indicate whether each material, system or method listed is relevant to your study. If you are not sure if a list item applies to your research, read the appropriate section before selecting a response.

## Materials & experimental systems

| n/a | Involved in the study |
|---|---|
| ☒ | ☐ Antibodies |
| ☒ | ☐ Eukaryotic cell lines |
| ☒ | ☐ Palaeontology and archaeology |
| ☒ | ☐ Animals and other organisms |
| ☒ | ☐ Clinical data |
| ☒ | ☐ Dual use research of concern |

## Methods

| n/a | Involved in the study |
|---|---|
| ☒ | ☐ ChIP-seq |
| ☒ | ☐ Flow cytometry |
| ☒ | ☐ MRI-based neuroimaging |

