## [Peer Review File · Nature Methods]

Comparing chromatin contact maps at scale: methods and insights

Corresponding Author: Dr Katherine Pollard

Version 0:

Decision Letter:

10th Jul 2023

Dear Professor Pollard,

Your Analysis, "Comparing chromatin contact maps at scale: methods and insights", has now been seen by 2 reviewers. As you will see from their comments below, although the reviewers find your work of potential interest, they have raised a number of concerns. We are interested in the possibility of publishing your paper in Nature Methods, but would like to consider your response to these concerns before we reach a final decision on publication.

We therefore invite you to extensively revise your manuscript to fully address all these concerns, including comparing more methods, demonstrating the advantage of the proposed new methods, and others.

Link Redacted

We hope to receive your revised paper within 3 months. If you cannot send it within this time, please let us know. In this event, we will still be happy to reconsider your paper at a later date so long as nothing similar has been accepted for publication at Nature Methods or published elsewhere.

OPEN SCIENCE REQUIREMENTS

REPORTING SUMMARY AND EDITORIAL POLICY CHECKLISTS

DATA AVAILABILITY

All novel DNA and RNA sequencing data, protein sequences, genetic polymorphisms, linked genotype and phenotype data, gene expression data, macromolecular structures, and proteomics data must be deposited in a publicly accessible database, and accession codes and associated hyperlinks must be provided in the "Data Availability" section.

CODE AVAILABILITY

Please include a "Code Availability" subsection in the Online Methods which details how your custom code is made available. Only in rare cases (where code is not central to the main conclusions of the paper) is the statement "available upon request" allowed (and reasons should be specified).

MATERIALS AVAILABILITY

ORCID

Sincerely,

Lin Tang, PhD
Senior Editor
Nature Methods

Reviewers' Comments:

Reviewer #1:

Remarks to the Author:

This work evaluates computational approaches for comparative analysis of Hi-C/Micro-C contact maps. The authors compile a set of existing metrics/scores/features used for Hi-C data analysis alongside a new one named triangle score and mostly qualitatively evaluate differences in these scores to assess their ability to capture differences either across two cell types or across simulated conditions. Their work aims to provide a reference guide together with a codebase to alleviate the issues in the field for comparative analysis, which is an important task. The work, however, falls short of being authoritative due to exclusion of majority of prior work in comparative analysis of Hi-C data and lack of quantitative definition of statistically significant differences. The scope and the set of evaluations has to be broadened significantly for this work to achieve what it proposes.

Major concerns

1. The classification of map-informed vs feature-informed methods is not clear to me. Why not have one category that focuses on all major features of contact maps (loops, TADs, stripes, compartments) and their comparative analysis and another that works directly on contact maps (i.e., map-informed). I do not think "overlap of functional genomics data" is a comparative method. Also, the contact probability decay is generally used as a genome-wide measurement and its use for local comparisons is not well justified. It is also not clear to me how these are computed locally and plotted like in Figure 3A.
2. Lack of comparative tools such as HiCCompare, HiCDC+, TADCompare, HiCexplorer, FIND, Selfish, diffHiC, dcHiC, HOMER, ZipHiC and so on. These are tools, although none are perfect, that are designed for comparative analysis and having a benchmark paper without including a fair number of them is not ideal. An exhaustive and comprehensive list of these tools can be found at: https://github.com/mdozmorov/HiC_tools#differential-analysis
2. Real-life relevance of 1Mb at a time analysis. It is not possible to know where certain features, such as TADs, start and end. Therefore, although convenient for the real data analysis here, doing the differential analysis 1Mb at a time does not capitulate what users will be doing when utilizing this codebase. This should at least be addressed in the limitations and ideally solved by doing genome-wide analysis. I understand this may be necessary for evaluation of simulations and that is fine.
3. As also mentioned by the authors in the limitations, use of only one tool for loop calling and one for TAD calling is extremely difficult to justify given the literature on the differences among these tools. At least three representatives should be used for each class and these could be identified from review articles comparing such tools. Or from the GitHub link provided above.
4. The intuition behind the triangle score (TS) and how it works is missing from the main text. This is important since this is regarded as a novel contribution. Also, TS is referred to as "1D track" whereas Figure 3A depicts it as a 2D and triangle-like score heatmap. This is confusing. The time-consuming nature of this score and lack of its clear benefits make it unclear whether this is a worthwhile addition to the toolset.
5. Results discussion of Figure 3A is completely done on the basis of visual inspection. Given the sensitivity of these scores/metrics to noise, it is not acceptable to say they "detect" changes or not just by looking at the signals with no quantification. A sentence like "Compared to other approaches, directionality index performed best in identifying focal changes, like the loss of loops (Fig. 3A iv)" is completely unsubstantiated, for instance. The following sentence is not supported by any data either.

Minor

- It is not clear what type of contact map normalization is applied and when. For instance, I was unable to find whether read depth normalization is applied which will make sense for some cases of evaluation but not for other (e.g., intensity). Some of this information can be briefly given in the main text to provide context.

- Similarly, the resolution of contact maps used are not clear aside from Akita which uses 2048bp. Please mention these in the main text.
- Figure 3B, color codes are missing as to what is orange and black. Also, it is not clear whether the black arcs are differential loops between ESC and HFF.
- The parameter tuning needed for Fig 3B is concerning but well known. More practical guidelines around this issue could have been useful but that is also highly dependent on data characteristics. My concern is that this part reiterates the need but does not go beyond it.
- Fig 4B and 4C are a bit confusing and indirect. I personally found Supp figs 4,5 and 7 much more informative and maybe some of this material can be brought forward.
- How are the representative examples are selected from different classes in Fig 4D,E?
- Fig 4Eii, the representative example has no difference yet eigenvector method seems to capture this. Can you clarify?
- I think there is either a typo or misinterpretation where the text says "Profile v is the most dissimilar." and the next sentence. This really is not the case. Did you mean iv instead of v?
- Fig 5E is referred to twice in caption, second should be F.
- How do authors decrease the resolution in Fig 5? It is not clear how comparison is performed after this.
- Table 1 is very busy and it is a bit confusing to have check and cross marks together.
- Other references that can be considered for inclusion are: https://link.springer.com/protocol/10.1007/978-1-0716-1390-0_4
<https://www.sciencedirect.com/science/article/pii/S2001037021001276>

Reviewer #2:

Remarks to the Author:

Comparing chromatin contact maps at scale: methods and insights by Gunsalus, McArthur et al.

This manuscript performs a thorough comparison of new and existing methods used to contrast changes in contact matrices from chromatin conformation capture experiments and proposes useful guidelines for their usage. Using *in silico* perturbations, the authors measure the consequences of tinkering with defined biological and technical sources of change. The main message of the manuscript is that each comparison method has its strengths and vulnerabilities which can be most effectively avoided by using multiple methods in parallel, for which they provide a codebase and guidelines.

The authors evaluate 8 commonly used and 3 new Hi-C comparison methods that they classify into 'basic', 'map-informed' and 'feature-informed' depending on the basis of the analysis (Figure 1). The authors then use each method on 1 Mb windows across the human genome comparing Micro-C maps of embryonic stem cells to those of fibroblasts. They find that so-called 'basic' methods prioritise different types of change such as those produced by structural rearrangements in the case of correlation-based methods or those produced by changes in the range of contact frequency in the case of mean squared error-based methods (Figure 2). The authors then test 'map-informed' and 'feature-informed' methods in their ability to detect a set of known changes in the underlying map such as boundary-, stripe- and focal contact- differences and find that they all detect such changes albeit with different sensitivity and requiring parameter selection (Figure 3). The authors then expand their disruption detection tests to 22,500 *in silico* CTCF motif insertions and deletions, as well as random sequence deletions, finding different degrees of overlaps in their top scoring regions, as well as events prioritised by all methods and events only detected by subsets of them (Figure 4). Finally, the authors challenge these methods against single, biologically meaningful changes and technical artifacts characterising their sensitivity to these variables (Figure 5). Based on these perturbations the authors issue a series of guidelines to analyse chromatin contact maps including the usage of a combination of methods of different type and provide a GitHub repository with implementations of the methods compared for ease of use.

Overall, the manuscript very interesting. The analysis benchmark, guidelines and code provide a valuable resource for the community. The manuscript clearly written and easy to follow (except for some points that require clarification – see below). As such, in my opinion, the manuscript will be of interest to a broad audience and useful to the field.

However, there are a few points that are unclear in the current version of the manuscript that need to be addressed in full before some of the observations can be properly interpreted that I will review in turn.

Major comments

1. The Github repository is meant to be used as a codebase for easy implementation of these methods for benchmarking. However, the code provided is not in a state that can be easily used. There is no README or guidance for how to use the resource. The code is difficult to follow and in its current form is unlikely to be widely implemented/used by the community. It might be more useful if each of the methods would be provided in a separate .py script properly documented.
2. The authors introduce three new methods to compare chromatin structure. However, the manuscript lacks the necessary level of detail to evaluate these novel methods. In particular, the triangle method needs to be formally explained and technically evaluated. Without a full formal evaluation of the method, including evaluations against proper biological perturbations, it is unclear how these methods compare. Furthermore, the authors find that the output of the three new methods that they introduce is comparable to existing methods. It is therefore unclear what is the advantage of the proposed methods, especially given their time and memory requirements. Could the authors provide specific examples of them outperforming existing methods or measurements of higher detection sensitivity or specificity to a specific set of changes?
3. While the manuscript focuses on changes happening in 1Mb windows, it is unclear how methods more sensitive to smaller size changes such as insulation and directionality index can be compared with methods specifically designed for bigger

window sizes. This brings into question whether the methods are compared in a situation where they would perform at their best. It would be useful if the authors could include such a comparison or explain why such a comparison is not relevant.

4. It is also unclear why the authors use 1 Mb windows and whether different (especially smaller) window sizes would affect the results. It would be interesting to see how the methods align when using smaller and bigger window sizes.

5. The reasoning behind the selection of methods compared in the manuscript is also not clear. There are other methods designed to detect differential interactions between Hi-C matrices which are routinely used in the field such as diffHic (Lun and Smyth, 2015) and methods to detect structure in Hi-C maps such as the Arrowhead algorithm which also considers the triangular nature of contact domains (Rao et al. 2014) and new 1D metrics to study Hi-C maps. (HiC1Dmetrics, Wang & Nakato 2022). It would be very useful to include some of these into the comparison.

6. In Figure 3, perhaps a compartment-level change could be included to illustrate the performance of these methods against such a change.

7. It is unclear if the matrix pre-processing that the authors perform influences the results presented in the manuscript. In particular, the logarithmic transformation and clipping of observed/expected normalised contacts, linear interpolation and convolution might affect the results of the algorithms tested in different ways. For example, insulation and directionality index are usually calculated on matrices without these transformations. This needs to be formally tested against the different methods.

Minor points

8. Genome coordinates as well as gene models are missing in several figures (2, 3a, 4e). These would be helpful for the biological interpretation of these panels.

9. If available, CTCF binding data for the wildtype matrix as well as annotations of the location of deleted/inserted motifs in the examples shown in Figure 4e would be useful to be able to interpret the observed changes.

10. It is unclear if the correlation matrix comparing the scores produced by different methods in Figure 4b is a Pearson correlation or a rank-based correlation and whether the differential distributions of the compared scores affect these correlations.

11. The in silico CTCF motif deletions and insertions can be split into those occurring in CTCF clusters, or at TAD boundaries, to properly interpret if a lack of a CTCF site would be expected to lead to the loss of a boundary. This is also related to point #9.

12. The authors mainly test these methods on Micro-C data. Would these findings translate to Hi-C data?

13. Would it be possible to measure the sensitivity and specificity of these methods to a set of known changes using a ROC curve? This would help when comparing the newly introduced methods to existing methods.

14. Figure 1 has a misaligned method in panel c.

15. Table 1 should include an evaluation of the need for parameter tuning necessary for these methods.

16. Please clarify if the insulation index window of 10 bins corresponds to 10 kb.

17. There is a typo (SSI instead of SSIM) in Supplementary Figure 7a, Supplementary Figure 8a and the codebase.

Version 1:

Decision Letter:

Our ref: NMETH-AS52176A

24th Sep 2024

Dear Dr. Pollard,

Thank you for submitting your revised manuscript "Comparing chromatin contact maps at scale: methods and insights" (NMETH-AS52176A). We very much apologize for the longer review time than we would hope. Your paper has now been seen by Reviewer 1 who finds that the paper has improved in revision (please see their comments below). Meanwhile, despite multiple efforts to contact Reviewer 2, we are not able to receive their review comments. In this case, we have invited Reviewer 1 to comment on your responses to Reviewer 2's concerns raised in the last round of review, who thinks all major comments are well addressed. In light of these review comments, we'll be happy in principle to publish it in Nature Methods, pending minor revisions to satisfy the referees' final requests and to comply with our editorial and formatting guidelines.

TRANSPARENT PEER REVIEW

ORCID

Sincerely,

Lin Tang, PhD
Senior Editor
Nature Methods

Reviewer #1 (Remarks to the Author):

I appreciate this very comprehensive work that will be valuable for the field. Also, the manuscript is now more focused and streamlined. The authors addressed my major concerns, few minor suggestions are below:

- chr names and coordinates should be indicated on the figures such as Figure 2 and Supp Fig 2
- supp table 1, column "Involves parameters" has to clarify whether this refers to input parameters or learned or tuned parameters
- Figure 5 and evaluation: the way that the simulated data is generated (i.e. creating new triangles) obviously favors a method that directly quantifies differences across triangles. At least this caveat should be acknowledged to not give the impression that Triangle method is the one and done type of solution for the real Hi-C data analysis, which will have more complex patterns and confounders.

Version 2:

Decision Letter:

14th Feb 2025

Dear Dr Pollard,

We very much appreciate your patience when we check the revised version of your Analysis, "Comparing chromatin contact maps at scale: methods and insights". I am pleased to inform you that this paper has now been accepted for publication in Nature Methods. The received and accepted dates will be 20th Apr 2023 and 14th Feb 2025. This note is intended to let you know what to expect from us over the next month or so, and to let you know where to address any further questions.

Over the next few weeks, your paper will be copyedited to ensure that it conforms to Nature Methods style. Once your paper is typeset, you will receive an email with a link to choose the appropriate publishing options for your paper and our Author Services team will be in touch regarding any additional information that may be required. It is extremely important that you let us know now whether you will be difficult to contact over the next month. If this is the case, we ask that you send us the contact information (email, phone and fax) of someone who will be able to check the proofs and deal with any last-minute problems.

Please feel free to contact me if you have questions about any of these points. Thank you very much for publishing your paper at Nature Methods!

Best regards,

Lin Tang, PhD
Senior Editor
Nature Methods

** Visit the Springer Nature Editorial and Publishing website at http://editorial-jobs.springernature.com?utm_source=ejP_NMeth_email&utm_medium=ejP_NMeth_email&utm_campaign=ejp_Nmeth for more information about our career opportunities. If you have any questions please click [here](mailto:editorial.publishing.jobs@springernature.com).**

Open Access This Peer Review File is licensed under a Creative Commons Attribution 4.0 International License, which permits use, sharing, adaptation, distribution and reproduction in any medium or format, as long as you give appropriate credit to the original author(s) and the source, provide a link to the Creative Commons license, and indicate if changes were made. In cases where reviewers are anonymous, credit should be given to 'Anonymous Referee' and the source. The images or other third party material in this Peer Review File are included in the article's Creative Commons license, unless indicated otherwise in a credit line to the material. If material is not included in the article's Creative Commons license and your intended use is not permitted by statutory regulation or exceeds the permitted use, you will need to obtain permission directly from the copyright holder.

Reviewer #1:

Remarks to the Author:

This work evaluates computational approaches for comparative analysis of Hi-C/Micro-C contact maps. The authors compile a set of existing metrics/scores/features used for Hi-C data analysis alongside a new one named triangle score and mostly qualitatively evaluate differences in these scores to assess their ability to capture differences either across two cell types or across simulated conditions. Their work aims to provide a reference guide together with a codebase to alleviate the issues in the field for comparative analysis, which is an important task. The work, however, falls short of being authoritative due to exclusion of majority of prior work in comparative analysis of Hi-C data and lack of quantitative definition of statistically significant differences. The scope and the set of evaluations has to be broadened significantly for this work to achieve what it proposes.

Major concerns

1. The classification of map-informed vs feature-informed methods is not clear to me. Why not have one category that focuses on all major features of contact maps (loops, TADs, stripes, compartments) and their comparative analysis and another that works directly on contact maps (i.e., map-informed). I do not think "overlap of functional genomics data" is a comparative method. Also, the contact probability decay is generally used as a genome-wide measurement and its use for local comparisons is not well justified. It is also not clear to me how these are computed locally and plotted like in Figure 3A.

We reclassified the existing and new methods into: (1) Global methods and (2) Contact Map methods. Global methods are statistical measures, like mean squared error and Spearman correlation, that compare the entire matrices and do not incorporate knowledge of 3D contact patterns. Contact map methods are specifically developed to compare contact map matrices. To remove any confusion on methods we adapted from existing contact map methods to compare matrices at specific loci, we renamed them as follows. Contact probability decay was renamed to Distance enrichment, since it focuses on changes between distal loci. Directionality index was renamed to Contact directionality since it identifies the directionality of contact concentration, for example towards the TAD at its boundary. We also added explanations for the motivation and adaptation of all four newly adapted methods in the Methods section. We also clarified in the Supplemental Text how Distance enrichment and Contact Directionality are computed. Furthermore, we removed "overlap of functional genomics data" from the manuscript given that it is not a method for quantitatively comparing maps but rather qualitatively evaluating differences in functionality.

2. Lack of comparative tools such as HiCCompare, HiCDC+, TADCompare, HiCexplorer, FIND, Selfish, diffHiC, dcHiC, HOMER, ZipHiC and so on. These are tools, although none are perfect, that are designed for comparative analysis and having a benchmark paper without including a fair number of them is not ideal. An exhaustive and comprehensive list of these tools can be found at:

https://github.com/mdozmorov/HiC_tools#differential-analysis

We agree additional comparative methods should be included. We have now benchmarked 12 additional methods: Arrowhead, CHESS, dcHiC (corr and mse), HiC1Dmetrics (CD, CIC, deltaDLR, ISC, and SSC), HiCcompare, Selfish, and TADcompare. In total, we were able to evaluate and compare 25 tools in total. We attempted to implement all of the tools suggested by both reviewers but for various reasons, enumerated in Supplemental Table 1, were not able to include all of them in our analyses. Though not exhaustive, the expanded set represents the diversity of strategies that have been proposed in the literature. We appreciate the reviewer's suggestion; it has expanded our benchmark notably.

We also note that many existing methods have limitations in the types of inputs they accept. This constrained our ability to include some methods in our analyses—most notably benchmarking analyses based on predicted maps. We attempted to manipulate predicted maps into formats accepted by existing tools, but this approach proved to require different assumptions and steps for different methods. For example, to run HiCcompare on predicted maps, we created a hic.table object using a sparse upper triangular format extracted from the predicted maps. The results of these comparisons were sensitive to choices made in transforming the predicted maps and often did not seem reasonable. Based on these tests, we concluded that many existing comparative tools are not compatible with predicted maps. For the purpose of this manuscript, we did not further evaluate the applicability of these tools on predicted maps. Since many tools cannot be applied to predicted maps, we focused the manuscript on experimental maps and kept our existing predicted map analyses in the supplement, to show generalization across more map types.

2. Real-life relevance of 1Mb at a time analysis. It is not possible to know where certain features, such as TADs, start and end. Therefore, although convenient for the real data analysis here, doing the differential analysis 1Mb at a time does not capitulate what users will be doing when utilizing this codebase. This should at least be addressed in the limitations and ideally solved by doing genome-wide analysis. I understand this may be necessary for evaluation of simulations and that is fine.

With our shift to emphasizing results from experimental maps, we were able to eliminate this limitation. For the full set of methods, we have evaluated performance on experimental maps at three different window sizes: 100 kb, 1 Mb, and 10 Mb. Since some of the methods are not easily applied on a chromosome-wide scale, we were unable to perform a comparable genome-wide analysis for all methods. We now mention this in the discussion. We also now point out that comparisons of the predicted maps are limited by the maximum input sequence size of the model.

3. As also mentioned by the authors in the limitations, use of only one tool for loop calling and one for TAD calling is extremely difficult to justify given the literature on the differences among these tools. At least three representatives should be used for each class and these could be identified from review articles comparing such tools. Or from the GitHub link provided above.

We recognize the value in evaluating multiple representatives of both TAD and loop callers. Among the 25 methods tested, Loops, HiCcompare, and Selfish focus on loops, while TADs, TADcompare and Arrowhead focus on TADs. We specifically evaluate how these compare in the Results section. Namely,

scores from loop-calling tools do not cluster together while those from TAD calling tools do. This provides comparative performance for multiple loop and TAD based methods.

4. The intuition behind the triangle score (TS) and how it works is missing from the main text. This is important since this is regarded as a novel contribution. Also, TS is referred to as "1D track" whereas Figure 3A depicts it as a 2D and triangle-like score heatmap. This is confusing. The time-consuming nature of this score and lack of its clear benefits make it unclear whether this is a worthwhile addition to the toolset.

We have expanded the explanation of the Triangle score (now named "Triangle") methodology, both in the Methods section under Adapted methods, and in the Supplemental Text under Method descriptions. We demonstrate through various analyses that both its concordance with established methods and its unique capabilities quantitatively justify the additional computational costs. It provides complementary information that aids biological interpretation.

Triangle is one of the top performing methods using our new simulation-based AUC-ROC and AUC-PR analyses (Figure 5D). Triangle (corr) and (mse) are resistant to noise (Figure 4A) and Triangle (corr) is resistant to resolution changes (Figure 4B), while most other methods do not have both these characteristics. Triangle (corr) is also able to identify map disruptions that other methods do not pick up in their top 5% highest scoring maps (Supplemental Figure 7). For these reasons, we believe that the additional computational cost for the Triangle method is outweighed by its unique and complementary capabilities. Therefore, it is a useful additional metric when one has sufficient computing to use it.

Additionally, we agree that the 2D representation of the method was confusing so we removed it and we no longer refer to it as a 1D track.

5. Results discussion of Figure 3A is completely done on the basis of visual inspection. Given the sensitivity of these scores/metrics to noise, it is not acceptable to say they "detect" changes or not just by looking at the signals with no quantification. A sentence like "Compared to other approaches, directionality index performed best in identifying focal changes, like the loss of loops (Fig. 3A iv)" is completely unsubstantiated, for instance. The following sentence is not supported by any data either.

We agree that quantitative comparisons are necessary to support the claims made; we added substantial new analyses to address this great suggestion. First, we clarified in the text that the qualitative observations from former Figure 3A are purely based on visual inspection. We have moved this figure to the supplement and used it merely as a way to visualize how these methods evaluate changes across the map. Second, we highlight the new and existing quantitative comparisons of the methods based on: clustering their scores when applied to representative contact maps (Figure 3, Supplemental Figure 12), simulation experiments to test the sensitivity of methods to different perturbations (Figure 4 and Supplemental Figure 19), direct comparison of overlaps between high scoring maps for each method (Supp Figure 8 and 18), and a new quantitative analysis on measuring method performance using ROC and Precision-Recall curves (Figure 5 and Supplemental Figure 20). The latter involves simulating a set of positive map pairs with structural changes (added or strengthened TAD boundary) and negative map pairs with technical changes (added noise or decreased resolution) and using these to evaluate how well

each method is able to correctly score the positive and negative set as such. Overall these quantitative analyses provide additional resolution with which to interpret strengths and weaknesses of each scoring method.

Minor

1. It is not clear what type of contact map normalization is applied and when. For instance, I was unable to find whether read depth normalization is applied which will make sense for some cases of evaluation but not for other (e.g., intensity). Some of this information can be briefly given in the main text to provide context.

Thank you for this recommendation. We added clarification on the normalization process applied to both experimental and predicted maps, which is a standard normalization process that was used to train the Akita model that computes predicted maps. In brief, we performed genome-wide iterative correction (ICE) normalization, adaptive coarse-graining, observed over expected normalization, log transformation, clipping to (-2,2), linear interpolation and convolution. The combination of ICE and observed over expected normalization helps eliminate biases and genomic distance-dependent decay within the sample and adjust sequence depth difference between samples. Other processing steps are helpful for prioritizing locus-specific patterns, addressing sparsity in the maps and maintaining consistency between experimental data and computational predictions. We now describe this in the Results section and point to new text in the Methods section under Experimental maps with the details. We also cite the Akita paper that includes more information on the normalization.

2. Similarly, the resolution of contact maps used are not clear aside from Akita which uses 2048bp. Please mention these in the main text.

We have added information on the resolution used in the main text, in figure captions when applicable, and in the Methods section.

3. Figure 3B, color codes are missing as to what is orange and black. Also, it is not clear whether the black arcs are differential loops between ESC and HFF.

In former Figure 3B, we removed the tracks and, instead, color-coded the markers on the maps based on whether those features are shared between the maps or unique to one map. We added a color code and explained this in the figure legend.

4. The parameter tuning needed for Fig 3B is concerning but well known. More practical guidelines around this issue could have been useful but that is also highly dependent on data characteristics. My concern is that this part reiterates the need but does not go beyond it.

While we agree that practical guidelines surrounding parameter selection would be very beneficial to users, this is challenging to evaluate broadly and needs to be fine-tuned to each application. We believe this is outside of the scope of this manuscript; our aim is to describe how different methods compare as

generally as possible. By pointing out which methods allow parameter tuning, we hope to show users that those methods provide both more flexibility and more complexity as parameter tuning requires an understanding of the types of changes to highlight and how conservative to be in finding them. To address this comment, we added text to clarify that methods with tuning parameters have potential to perform better than reported on specific applications with adjustment of the parameters.

5. Fig 4B and 4C are a bit confusing and indirect. I personally found Supp figs 4,5 and 7 much more informative and maybe some of this material can be brought forward.

We updated Figure 4B and C to integrate the new experimental data analyses. The goal of these panels is to summarize trends and show which methods are similar and different to each other; former Supplementary Figures 4, 5, and 7 do not address this direct comparison. Nonetheless, we agree that former Supplementary Figure 4, 5, and 7 provide complementary information, and we are glad that the reviewer found them to be informative. The message portrayed in former Supplementary Figure 4 is similar to that in the updated Figure 2; former Supplementary Figure 5 provides useful examples, but does not show a conclusive result enough to be a main figure; and former Supplementary Figure 7 overlaps with the information in Figure 3. Thus, given space constraints, we respectfully chose to keep these as supplementary figures (renumbered now), while discussing them fully in the main text. We also moved former Figure 3 to the supplement.

6. How are the representative examples are selected from different classes in Fig 4D,E?

The representative map examples highlighted in former Figure 4D-E were manually selected by visual inspection of the pattern of scores (grayscale heatmap). We chose examples whose scoring patterns most closely matched the pattern in each section of the heatmap. This selection approach is now explicitly stated in the caption of the figure, now Supplementary Figure 12. Since portraying and explaining these examples was not clear, we did not pull out representative maps for the new version of former Figure 4 (now Figure 3), which is based on experimental data. We are grateful for this feedback.

7. Fig 4Eii, the representative example has no difference yet eigenvector method seems to capture this. Can you clarify?

The representative example in Fig 4Eii only shows slight change in strength at the boundary, which scores low with all methods other than Eigenvector. We mention this in the figure caption, suggesting that the method can be highly sensitive to such changes.

8. I think there is either a typo or misinterpretation where the text says "Profile v is the most dissimilar." and the next sentence. This really is not the case. Did you mean iv instead of v?

Thank you for catching the incorrect text reference to profile v, which should have referred to profile iv. We have since removed that section of the text since former Figure 4 was moved to Supplemental Figure 12.

9. Fig 5E is referred to twice in caption, second should be F.

We corrected the duplicate use of former Figure 5E, now Figure 4, in the caption which now properly refers to Figures 4E and 4F. Thank you.

10. How do authors decrease the resolution in Fig 5? It is not clear how comparison is performed after this.

Thanks for pointing this out. We have added text clarifying how the binning to lower resolutions was performed in the Methods section under Simulated maps. In brief, we averaged values in adjacent bins to achieve the target number of bins. After this, the comparison is performed between two matrices of different bin sizes with no adjustments to comparison methods necessary.

11. Table 1 is very busy and it is a bit confusing to have check and cross marks together.

Thank you for the helpful feedback. We agree and have redesigned Table 1 to simplify the formatting by removing the potentially confusing check/cross notation and only having black checkmarks, either single or double denoting the strength of the feature. The lack of a checkmark denotes the lack of that feature. This is now explained in the table caption.

12. Other references that can be considered for inclusion are:

https://link.springer.com/protocol/10.1007/978-1-0716-1390-0_4

<https://www.sciencedirect.com/science/article/pii/S2001037021001276>

Thank you, we have added the suggested references in the Introduction section.

Reviewer #2:

Remarks to the Author:

Comparing chromatin contact maps at scale: methods and insights by Gunsalus, McArthur et al.

This manuscript performs a thorough comparison of new and existing methods used to contrast changes in contact matrices from chromatin conformation capture experiments and proposes useful guidelines for their usage. Using *in silico* perturbations, the authors measure the consequences of tinkering with defined biological and technical sources of change. The main message of the manuscript is that each comparison method has its strengths and vulnerabilities which can be most effectively avoided by using multiple methods in parallel, for which they provide a codebase and guidelines.

The authors evaluate 8 commonly used and 3 new Hi-C comparison methods that they classify into ‘basic’, ‘map-informed’ and ‘feature-informed’ depending on the basis of the analysis (Figure 1). The authors then use each method on 1 Mb windows across the human genome comparing Micro-C maps of embryonic stem cells to those of fibroblasts. They find that so-called ‘basic’ methods prioritise different types of change such as those produced by structural rearrangements in the case of correlation-based methods or those produced by changes in the range of contact frequency in the case of mean squared error-based methods (Figure 2). The authors then test ‘map-informed’ and ‘feature-informed’ methods in their ability to detect a set of known changes in the underlying map such as boundary-, stripe- and focal contact- differences and find that they all detect such changes albeit with different sensitivity and requiring parameter selection (Figure 3). The authors then expand their disruption detection tests to 22,500 *in silico* CTCF motif insertions and deletions, as well as random sequence deletions, finding different degrees of overlaps in their top scoring regions, as well as events prioritised by all methods and events only detected by subsets of them (Figure 4). Finally, the authors challenge these methods against single, biologically meaningful changes and technical artifacts characterising their sensitivity to these variables (Figure 5). Based on these perturbations the authors issue a series of guidelines to analyse chromatin contact maps including the usage of a combination of methods of different type and provide a GitHub repository with implementations of the methods compared for ease of use.

Overall, the manuscript very interesting. The analysis benchmark, guidelines and code provide a valuable resource for the community. The manuscript clearly written and easy to follow (except for some points that require clarification – see below). As such, in my opinion, the manuscript will be of interest to a broad audience and useful to the field.

However, there are a few points that are unclear in the current version of the manuscript that need to be addressed in full before some of the observations can be properly interpreted that I will review in turn.

Major comments

1. The Github repository is meant to be used as a codebase for easy implementation of these methods for benchmarking. However, the code provided is not in a state that can be easily used. There is no README or guidance for how to use the resource. The code is difficult to follow and in its current form is unlikely

to be widely implemented/used by the community. It might be more useful if each of the methods would be provided in a separate .py script properly documented.

We fully agree that usability is critical for the codebase to achieve impact as a resource. As suggested, we restructured the code base to make it more digestible and user friendly. We separated each scoring method into its own script or combined it with other methods if they share functions, have the same inputs, and are written in the same language. We added a README overview outlining the available scripts as well as a brief description of how to use each method individually or how to run multiple methods together. We have also enhanced code readability and added extensive method docstrings. We believe that these changes will enable others to build on our codebase to apply existing methods to new datasets and benchmark new comparison methods.

2. The authors introduce three new methods to compare chromatin structure. However, the manuscript lacks the necessary level of detail to evaluate these novel methods. In particular, the triangle method needs to be formally explained and technically evaluated. Without a full formal evaluation of the method, including evaluations against proper biological perturbations, it is unclear how these methods compare. Furthermore, the authors find that the output of the three new methods that they introduce is comparable to existing methods. It is therefore unclear what is the advantage of the proposed methods, especially given their time and memory requirements. Could the authors provide specific examples of them outperforming existing methods or measurements of higher detection sensitivity or specificity to a specific set of changes?

We agree with the reviewer that further evaluation of the new methods is needed. To address this comment, we performed an array of new analyses that all together support the inclusion of the new methods. (We also note that in response to Reviewer #1, these new methods—Eigenvector, Contact directionality, Distance enrichment, and Triangle—are now referred to as adapted methods.) First, we tested all existing and adapted methods on scoring changes between Micro-C and Hi-C datasets from two different cell lines, at regions around differentially expressed genes. We explain this new analysis in Methods and show the resulting data in the new Figure 3. Additionally, we quantitatively evaluate how all methods perform using area under the curve (AUC) of the receiver operating characteristic (ROC) and precision-recall curves (analysis explained in Methods). These comparisons show that performance for the adapted methods is comparable to existing methods, with Triangle outperforming most methods. Additionally, we support the utility of adapted methods, like Triangle, by showing that while they generally agree with other methods, they uniquely identify maps with biologically meaningful differences that are not scored highly by other methods. Throughout the text, we have specifically highlighted how the adapted methods perform and why their inclusion is justified. Additionally, we further explained the motivation for incorporating these adapted methods in the main text in the Methods section. Taking all of the above analyses into account, we conclude that the new methods provide additional utility to existing methods.

3. While the manuscript focuses on changes happening in 1Mb windows, it is unclear how methods more sensitive to smaller size changes such as insulation and directionality index can be compared with methods specifically designed for bigger window sizes. This brings into question whether the methods are

compared in a situation where they would perform at their best. It would be useful if the authors could include such a comparison or explain why such a comparison is not relevant.

To address the scale issue, we have performed complementary analyses evaluating both 100 kb and 10 Mb windows, in addition to the original 1 Mb analysis, all centered at the same regions. To compare how methods compare to each other at different scales, we visualize their clustering (Supplemental Figure 10). Reassuringly, we found the overall relationship between methods to be highly consistent across window sizes. Although most methods group together across scales, a few do not. We discuss this in the Results text and figure caption.

4. It is also unclear why the authors use 1 Mb windows and whether different (especially smaller) window sizes would affect the results. It would be interesting to see how the methods align when using smaller and bigger window sizes.

We used the 1 Mb window in the original manuscript because it roughly reflects the scale at which TADs and the biologically relevant contact patterns within them form. Moreover, it is the window size that the Akita model takes as input. In the revised manuscript, we have transitioned most figures to experimental data which allows us to compare various window sizes. As suggested, we have compared how the methods align when using smaller and larger window sizes, as explained in the response to the comment above.

5. The reasoning behind the selection of methods compared in the manuscript is also not clear. There are other methods designed to detect differential interactions between Hi-C matrices which are routinely used in the field such as diffHiC (Lun and Smyth, 2015) and methods to detect structure in Hi-C maps such as the Arrowhead algorithm which also considers the triangular nature of contact domains (Rao et al. 2014) and new 1D metrics to study Hi-C maps. (HiC1Dmetrics, Wang & Nakato 2022). It would be very useful to include some of these into the comparison.

We agree with the reviewer that our original list of methods was not comprehensive. To overcome this, we evaluated 12 additional methods, including ones suggested by the reviewer here: Arrowhead, CHES, dcHiC (2 implementations), HiC1D metrics (5 implementations), HiCcompare, Selfish, and TADCompare. Unfortunately, diffHiC requires replicates while most other methods cannot handle them. Since this difference in input data would make direct comparison challenging, we did not include diffHiC in the analysis. We now explain why certain commonly used methods were not included in the manuscript in Supplementary Table 1. Nonetheless, these methods are still cited and described.

6. In Figure 3, perhaps a compartment-level change could be included to illustrate the performance of these methods against such a change.

To address the fact that not all types of changes of interest are captured at the 1 Mb scale, we added comparison of how the methods perform at larger and smaller scales in Supplementary Figure 10. Please also see response to major reviewer comment 3, where we describe our new analyses and results comparing scores across different window sizes (Supplemental Fig. 10). We appreciate the suggestion to include different compartment-level changes in Figure 3, but since there were not dramatic differences in

how methods cluster across different window sizes, we chose to keep the message in that figure focused on the comparison of different methods and genomic features that contribute to their scores.

7. It is unclear if the matrix pre-processing that the authors perform influences the results presented in the manuscript. In particular, the logarithmic transformation and clipping of observed/expected normalised contacts, linear interpolation and convolution might affect the results of the algorithms tested in different ways. For example, insulation and directionality index are usually calculated on matrices without these transformations. This needs to be formally tested against the different methods.

We agree that the effects of various normalization steps should be evaluated in our benchmark. Therefore, we applied each scoring method to experimental maps with differentially expressed genes between two cell types (H1hESC versus HFFc6, as in the primary analyses), but skipping each of the steps in the normalization process. We compared the resulting scores to the scores from the fully normalized data and found that most steps have a minimal impact on scores, although the log transformation steps does affect a subset of methods. As expected, correlation based methods are robust to various monotone transformations associated with normalization. These trends, along with several examples, are shown in a new figure, Supplementary Figure 14.

Since Akita was trained on normalized maps, we cannot compare predicted maps without normalization. Thus, given that most normalization steps do not have a large impact and to keep our analyses consistent between predicted and experimental maps, we focus in the main text on normalized experimental maps. Furthermore, since we consider multiple conditions across methods—namely different window sizes, different resolutions, different experiments (Hi-C and Micro-C), and different map types (predicted vs. experimental)—considering multiple different methods for pre-processing for each combination of other variables would lead to a combinatorial explosion in the evaluation .

Minor points

8. Genome coordinates as well as gene models are missing in several figures (2, 3a, 4e). These would be helpful for the biological interpretation of these panels.

We thank the reviewer for this suggestion. We added gene tracks to Figure 2 and genome coordinates to the figure caption. We have moved previous Figures 3 and 4 to the supplemental material.

9. If available, CTCF binding data for the wildtype matrix as well as annotations of the location of deleted/inserted motifs in the examples shown in Figure 4e would be useful to be able to interpret the observed changes.

We agree, but in the revised manuscript we now focus on experimental data and have moved former Figure 4 to the Supplementary Material.

10. It is unclear if the correlation matrix comparing the scores produced by different methods in Figure 4b is a Pearson correlation or a rank-based correlation and whether the differential distributions of the compared scores affect these correlations.

We have clarified in the caption that former Figure 4b shows Pearson correlation coefficients between methods. This is one of many clustering methods that we use to evaluate how the different methods compare to one another—others include PCA and Ward’s clustering with Euclidean distance. For simplicity, we have only included this one type of correlation in former panel 4b, with results from other measures of similarity in other figure panels.

11. The in silico CTCF motif deletions and insertions can be split into those occurring in CTCF clusters, or at TAD boundaries, to properly interpret if a lack of a CTCF site would be expected to lead to the loss of a boundary. This is also related to point #9.

Now that the focus of the manuscript is on experimental maps, the CTCF insertion and deletions merely serve as a method to easily generate maps that are different from each other in various ways in order to compare methods on predicted maps in addition to the experimental ones. Thus, while this analysis would be interesting, we think that it is outside the scope of the revised manuscript. We have therefore changed the wording to focus on how methods compare across diverse maps, rather than attempting to interpret how methods score CTCF deletions and insertions differently.

12. The authors mainly test these methods on Micro-C data. Would these findings translate to Hi-C data?

We agree that testing on Hi-C datasets is critical to demonstrate the generalizability of these methods beyond Micro-C data. Therefore, we performed extensive additional analyses applying the same computational pipeline to deeply sequenced Hi-C datasets from matched cell types. We then compared scores across all methods between the two experiment types (Supplemental Figure 9 and 11). We now discuss when results from the two technologies are highly correlated and when they are not in the figure caption, as well as the main text. Briefly, we find that scores from Hi-C and Micro-C data correlate well overall and we see similar patterns of how different resolution maps compare between Hi-C and Micro-C data.

13. Would it be possible to measure the sensitivity and specificity of these methods to a set of known changes using a ROC curve? This would help when comparing the newly introduced methods to existing methods.

We thank the reviewer for this suggestion. We measured the specificity and sensitivity of all the methods that can be applied to predicted maps using area under the ROC and precision-recall curves. For this analysis, the positives were simulated map pairs with structural changes (added or strengthened TAD boundary) and the negatives were simulated map pairs with technical changes (increased noise or decreased resolution). We added a new Figure 5 showing these results, an interpretation of the results in the main text, a summary of these results and the other simulation results in Table 1, as well as the process for this analysis in the Methods section. We thank the reviewer for suggesting this great addition to the benchmark.

14. Figure 1 has a misaligned method in panel c.

Thank you for catching the misaligned method in Figure 1c. We have corrected this figure to properly align all method labels with their corresponding data points.

15. Table 1 should include an evaluation of the need for parameter tuning necessary for these methods.

As suggested, we added text in Supplemental Table 1 noting which methods require extensive parameter tuning or optimization versus which methods can be readily applied with default settings. This information allows readers to better assess the practicalities and opportunities associated with each tool.

16. Please clarify if the insulation index window of 10 bins corresponds to 10 kb.

Thanks for the suggestion. The 10-bin insulation index window refers to Akita bins which are 2,048 bp each. We have clarified this in the Methods section.

17. There is a typo (SSI instead of SSIM) in Supplementary Figure 7a, Supplementary Figure 8a and the codebase.

Thank you for correcting the SSIM typos. We have reviewed and corrected all instances of "SSI" to the proper "SSIM" terminology throughout the manuscript, supplementary figures, and legends.

Reviewer #1:

Remarks to the Author:

I appreciate this very comprehensive work that will be valuable for the field. Also, the manuscript is now more focused and streamlined. The authors addressed my major concerns, few minor suggestions are below:

- chr names and coordinates should be indicated on the figures such as Figure 2 and Supp Fig 2

We moved chromosome names and coordinates from the Figure 2 caption to the figure itself, and added chromosome names and coordinates to Supplementary Fig. 2.

- supp table 1, column "Involves parameters" has to clarify whether this refers to input parameters or learned or tuned parameters

We changed the column name to "Involves parameter tuning" to clarify that we are referring to input parameters that should be tuned for desired application and performance.

- Figure 5 and evaluation: the way that the simulated data is generated (i.e. creating new triangles) obviously favors a method that directly quantifies differences across triangles. At least this caveat should be acknowledged to not give the impression that Triangle method is the one and done type of solution for the real Hi-C data analysis, which will have more complex patterns and confounders.

We agree that our positive map pairs might be biased towards certain types of map changes, such as splitting up TADs and introducing two triangle shapes. To address this, we note this caveat in the Results section where we describe the Figure 5 simulation experiment.